# DETECTING PROBLEMATIC QUESTIONS TO SUPPORT MATH WORD PROBLEM DESIGN

## ABSTRACT

When designing math word problems, teachers must ensure the clarity and precision of the question to avoid multiple interpretations and unanswerable situations, thereby maintaining consistent grading standards and effectiveness. We address these issues to provide comprehensive support to teachers in creating clear, solvable, and formal math word problems. In this paper, we present MathError, a dataset of real-world math word problems annotated with error types to investigate the need for question correction. Our work explores how large language models (LLMs) can assist teachers in detecting problematic questions to support math word problem design in scenarios with limited data, simulating real-world conditions with minimal training samples. Preliminary results demonstrate the models' capabilities in detecting problematic questions and identify areas for further research and development in educational applications.

## 1 INTRODUCTION

When coming up with exam questions, teachers must ensure that the questions are clear and precise. This prevents students from misunderstanding the questions, which could lead to inconsistent grading standards and render the questions ineffective. This is particularly important for math word problems where there should usually be only one correct answer. However, when questions are formulated, there may be blind spots or minor oversights that lead to misinterpretations by students, or a lack of crucial details, making immediate comprehension difficult. For example, "The original price of an apple is 2 dollars. It has been discounted twice: the first discount is 10%, and the second discount is 5%. What is the current price of the apple?" It is unclear whether the second discount is to be applied to the original price or to the price after the first discount. Consequently, this description may cause confusion and uncertainty for students. It is therefore essential to construct a system that assists teachers in ensuring that the questions do not have multiple interpretations or are unanswerable.

Several studies have investigated situations where questions are unanswerable. Questions can be unanswerable in the following scenarios: (1) The knowledge sources are incomplete, failing to cover all the necessary facts required to answer the question (Patidar et al., 2023); (2) User-generated questions are poorly formatted, are missing entities or predicates, or contain ungrammatical phrases (Faustini et al., 2023); (3) The question is ambiguous and thus allows for more than one interpretation (Min et al., 2020); (4) Details in the question are inconsistent with the facts (Yen et al., 2021). Unlike previous studies that focus on knowledge base question answering or open-domain questions, we address multiple interpretations and unanswerable issues in math word problems. Specifically, we seek to detect the following conditions to support teachers in designing math word problems.

- Questions may result in multiple or unintended solutions due to imprecise descriptions, missing conditions or constraints, or unclear relationships between multiple values.
- Questions may be unanswerable as they contain unclear terms or noticeable omissions.

This issue could be influenced by language, as imprecise descriptions, missing conditions, and unclear relationships between values may manifest differently across languages. Additionally, the complexity of error types can vary depending on the difficulty level of the math problems. In this pilot study, we focus on elementary-level math word problems presented in Chinese.

Sun et al. (2024) have explored similar issues, emphasizing the importance of detecting problematic math word problems by constructing a dataset of unanswerable questions with predefined error types. This highlights the growing attention to the task of identifying and addressing challenges in math word problem design. However, while their study focuses on error types specifically created for their experiment, our work shifts towards detecting naturally occurring errors in real math word problems. In addition, we investigate more nuanced issues, such as multiple interpretations or unintended solutions, which can be challenging for models to identify. Hence, we extend the existing Chinese math word problem dataset—Math23K (Wang et al., 2017)—with error type annotations. Math23K questions provide rich textual descriptions that naturally meet our requirements.

Given the remarkable capabilities of large language models (LLMs) in language understanding and generation, recent studies use LLMs to generate test questions for student practice (Gonzalez et al., 2023; Feng et al., 2024; Song et al., 2023). The pedagogical ability of LLMs in mathematics education has also been studied (Yen & Hsu, 2023; Wang et al., 2024; Daheim et al., 2024). Some works have investigated the role of LLMs in assisting teachers with tasks such as distractor generation for math multiple-choice questions (Feng et al., 2024; Lee et al., 2024; Hunter McNichols et al., 2024). Liu et al. (2023) have shown that modeling ambiguity remains a significant challenge for LLMs, reinforcing the importance of developing methods to detect and address these problems. However, the capability of LLMs to recognize errors in math word problems and disentangle potential meanings is rarely explored. Thus, this work explores the capability of LLMs in identifying problematic math word problems. We further investigate a self-optimizing approach that allows the model to learn from its mistakes. By iteratively reflecting on the wrong predictions, the model refines instructions and demonstrations within the prompt, improving performance in detecting error types.

To sum up, the contributions of our work are threefold: (1) We assist teachers in ensuring the clarity of math word problems by detecting errors in question statements that can lead to several interpretations or render the problems unanswerable. (2) We present the MathError dataset,[1] which is designed for detecting errors in the statements of math word problems, to facilitate the investigation of the need for correcting problematic questions. (3) We explore a self-optimizing framework where the model iteratively refines its instructions and demonstrations through a reflection mechanism. This approach simulates real-world scenarios where data is scarce by utilizing only a few examples, offering a preliminary solution to the challenge of error detection in math word problems. Experimental results show that the prompts refined by our reflection mechanism yield better performance.

## 2 RELATED WORK

**Ambiguous and Unanswerable Questions:** There are several types of ambiguity: lexical, syntactic, semantic, pragmatic, and anaphoric (Li et al., 2024). Numerous works address disambiguation using methods such as syntactic and semantic parsing (Tanaka et al., 2007; Koller et al., 2008) or coreference resolution (Kocijan et al., 2019). In question-answering applications, ambiguous user queries lead to unanswerable queries. Methods have been developed to identify question answerability (Zhang et al., 2021; Yang et al., 2019) and generate clarification questions (Zamani et al., 2020; Krasheninnikov et al., 2022) or correct unanswerable questions (Yen et al., 2021). There has also been growing interest in addressing ambiguity in math word problems. Sun et al. (2024) define five different categories of unanswerable questions. Curated annotators modified answerable questions into unanswerable ones based on the categories. By contrast, we construct a dataset by annotating error types of real-world questions rather than modifying questions into unanswerable forms based on specific categories, which may make it difficult for models to identify specific patterns to determine whether a question contains errors. Consequently, our dataset comprises not only unanswerable questions but also questions with multiple possible solutions.

**Self-Optimization with LLMs:** LLMs have made significant advancements in producing coherent text and following given instructions (Wei et al., 2022a; Ouyang et al., 2022). Recently, methods have been investigated that elicit feedback from LLMs on self-generated solutions, enabling iterative improvement of outputs based on the feedback. Madaan et al. (2024) propose a framework that iteratively refines the generated output via self-evaluation. Several studies explore the use of LLMs for optimizing prompts. Zhou et al. (2022) employ the LLM to create instructions, select the proper instructions based on accuracy, and instruct the LLM to generate a semantically similar variant.

---

[1]The dataset and code will be released upon acceptance.

Pryzant et al. (2023) propose an approach to guide the LLM to provide textual feedback on how to revise an existing instruction at each step. Methods have also been developed to use natural language feedback generated by LLMs to refine the model's output (Chen et al., 2023; Ganguli et al., 2023; Shinn et al., 2023). Inspired by these studies, we introduce a reflection mechanism to our framework to refine the LLM's prompts in detecting problematic questions in mathematics.

## 3 FROM MATH23K TO MATHERROR

The two conditions outlined in Section 1 lead to errors that could significantly impact the clarity and accuracy of math word problems:

1. Multiple Interpretations (*INTPN*): The question allows for multiple possible interpretations, leading to more than one possible solution.
2. Informal Wording (*Informal*): The wording of the question is not formal or is incomplete, such as including unnecessary words or symbols, having noticeable omissions, or containing typographical errors, making the problem statement difficult to understand.
3. Unitless (*Unit*): The question does not specify the required unit, which may lead to confusion about what measurement is expected.
4. Unclear Relationship (*Rel*): The description fails to clearly indicate the relationship between the values, leading to misunderstandings about the question's meaning.
5. Calculation Error (*Calc*): The problem uses imprecise words to describe a mathematical expression, for instance, making it unclear whether to perform multiplication or division before addition or subtraction; this can cause students to calculate in the wrong order.

If a math word problem exhibits none of the issues mentioned above, it belongs to the *None* type. Note that we focus extends beyond detecting ambiguities in math word problems. We aim to address a challenge: ensuring that problem descriptions are formal and complete for use in official examinations. Our goal is to support teachers in refining the clarity and precision of problem statements, helping to eliminate informal language and incomplete details that could lead to misinterpretations. As math word problem error types can be highly diverse, it is difficult to immediately identify all possible types. The error type definitions and dataset construction are in Section 4.

## 4 DATASET CONSTRUCTION

**Error Type Definition**. Math23K comprises a total of 23,162 Chinese math word problems. To establish an initial set of error types, we randomly sampled 200 questions, referred to as the initial set, and categorized the errors present in these questions. We conducted a preliminary annotation of problematic descriptions, after which we consolidated these initial error types by merging similar ones. This process resulted in the identification and definition of five distinct error types. Yet, we are unsure whether the five error types are sufficient and whether they cover all possible errors. Additionally, we cannot guarantee the completeness of these error definitions. Thus, we established an iterative refinement annotation process to ensure the quality of the dataset annotation.

**Iterative Refinement Annotation**. We invited three annotators and split the entire dataset into four parts, with each person responsible for the labeling of 5,815 samples.[2] We labeled one of these parts by ourselves. To ensure consistency and quality across the dataset, we conducted a quantitative evaluation of the annotators' labeling accuracy using our pre-annotated set of 100 samples, which we refer to as the golden set. Of these, 23 samples contained math word problems with error statements, whereas 77 samples had no errors. To evaluate the correctness of the annotators' labels, we divided the 100 samples from the golden set into five subsets, each containing 20 samples, ensuring a balanced representation of error types. These golden sets were inserted into the subsets that the annotators were to label.

The annotation process was as follows. The annotators first labeled the initial subset of 20 samples from the golden set. These 20 samples were used to verify the annotators' labeling correctness. Since these samples had gold labels, we assessed the macro F-score of each annotator's results. Based on these results, we then provided further clarification and discussion on unclear aspects. Annotators are allowed to revisit and modify their previously labeled data if necessary. At this stage,

---

[2]Additional data annotation details are in Appendix A.

Table 1: Annotation agreement across five stages

|  | Stage 1 | Stage 2 | Stage 3 | Stage 4 | Stage 5 | Overall |
|---|---|---|---|---|---|---|
| Precision | 0.4546 | 0.7692 | 0.8335 | 0.9286 | 1.0000 | 0.7972 |
| Recall | 0.4166 | 0.4167 | 0.5556 | 0.5417 | 0.6000 | 0.5061 |
| F-score | 0.4348 | 0.5405 | 0.6667 | 0.6842 | 0.7500 | 0.6153 |

Table 2: Error types in math word problems

| Error type | Question | Reason |
|---|---|---|
| Multiple Interpretations | 一款原价3000元的空调先后两次降价，第一次降价10%，第二次又降价5%. 现在这款空调的售价多少元？(An air conditioner originally priced at 3000 yuan was discounted twice: the first discount was 10%, and the second discount was 5%. What is the current price of the air conditioner?) | The question does not clarify whether the second discount applies to the new price after the first discount, or to the original price. |
| Informal Wording | 某书店购书一律0.95，小红买了一本书，比原价便宜6元，这本书原价多少元？(A certain bookstore charges 0.95 for books purchased. A girl bought a book and saved 6 yuan off the original price. What was the original price of the book?) | The problem statement might lead one to believe that all books in the bookstore are priced at 0.95 yuan. The intended meaning, however, is that a 5% discount is applied to the original price. |
| Unitless | 一根木头，用32秒的时间分成了5段，以同样的速度将另一根木头分成7段，需要多少时间？(A piece of wood is cut into 5 sections in 32 seconds. At the same rate, how much time is needed to cut another piece of wood into 7 sections?) | The unit for the required time is not specified. For example, the question should ask, "How many seconds?" |
| Unclear Relationship | 4平家电商场现有各种品牌的电视机240台，比电冰箱多(1/5)，商场现有电冰箱多少台？(An appliance store currently has 240 televisions of various brands, which is more than refrigerators by 1/5. How many refrigerators does the store have?) | The question does not clarify the actual meaning of "more by 1/5". It should describe that the number of televisions is 1/5 times more than the number of refrigerators. |
| Calculation Error | 6000/59与35的差，商？(What is the quotient of the difference between 6000/59 and 35?) | The statement is incorrect because it does not clearly define the operation, leading to confusion. A more precise wording would be "6000 divided by 59, and then subtract 35 from this result." |
| None | 为灾区捐款，小华捐4.2元，比小丽多捐了0.4元，小华比小丽多捐几分之几？(Xiaohua donated 4.2 yuan, which is 0.4 yuan more than what Xiaoli donated. By what fraction did Xiaohua donate more than Xiaoli?) | The question is clear and correct. |

they could also provide feedback and suggest adjustments to the error type definitions. After confirming no immediate issues with the annotators' task, they continued labeling the next 20 samples from the golden set. This process was repeated using our pre-annotated gold labels to evaluate the annotators' performance for the 20 samples and identify discrepancies. This iterative process continued until all data in the golden set was annotated.

Table 1 presents the annotators' labeling correctness, evaluated using macro F-scores against the gold labels. The five stages correspond to macro F1 scores for subsets labeled at different stages, with increasing scores reflecting improved consistency and accuracy from iterative discussions and revisions. The inter-annotator agreement, as measured by Fleiss' kappa value, is 0.6038, indicating a moderate level of agreement. To ensure that annotators fully understood the task and applied consistent standards, we had them annotate the remaining 100 samples from the initial set, of which 21 were problematic math questions. The inter-annotator agreement for this round reached a Fleiss' kappa of 0.8103, representing substantial agreement. After ensuring a sufficient level of consistency in the annotation standards, we assigned the annotators to label their respective portions of the data.

**Dataset Analysis**. Table 2 presents the examples for each error type. This process resulted in 23,162 math word problems, with an error-type distribution of *Multiple Interpretations*, *Informal Wording*, *Unitless*, *Unclear Relationship*, *Calculation Error*, and *None* errors of 136, 1,076, 416, 606, 67, and 20,916, respectively. Although "Multiple Interpretation," "Informal Wording," and "Unclear Relationship" all stem from imprecise or incomplete problem descriptions, their effects differ. "Multiple Interpretation" may lead to various interpretations, resulting in multiple possible solutions. On the other hand, "Informal Wording" and "Unclear Relationship" involve cases where the teacher's intended question can still be inferred, but the phrasing is informal and unsuitable for offical examinations. "Unclear Relationship" specifically refers to situations where numerical relationships are not clearly defined, affecting the clarity of the problem. As for "Calculation Error," it typically arises from mixed mathematical and verbal expressions. For instance, in the example provided in the Table 2, a more precise wording would be, "6000 divided by 59, and then subtract

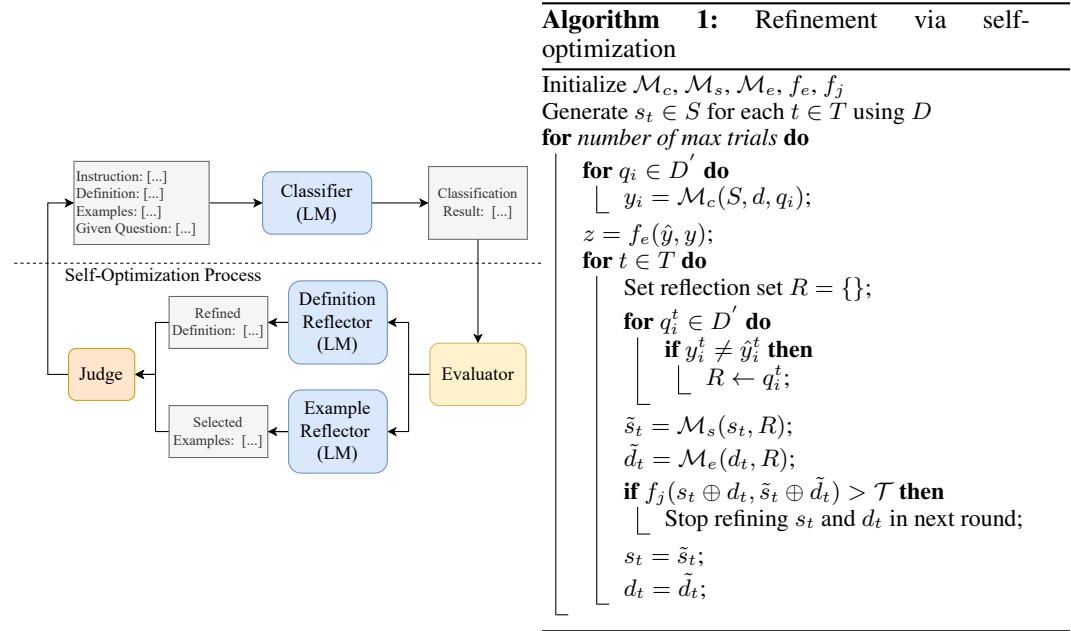

**Algorithm 1:** Refinement via self-optimization

---

Initialize $\mathcal{M}_c, \mathcal{M}_s, \mathcal{M}_e, f_e, f_j$
Generate $s_t \in S$ for each $t \in T$ using $D$
**for** *number of max trials* **do**
    **for** $q_i \in D'$ **do**
        $y_i = \mathcal{M}_c(S, d, q_i)$;
    $z = f_e(\hat{y}, y)$;
    **for** $t \in T$ **do**
        Set reflection set $R = \{\}$;
        **for** $q_i^t \in D'$ **do**
            **if** $y_i^t \neq \hat{y}_i^t$ **then**
                $R \leftarrow q_i^t$;
        $\tilde{s}_t = \mathcal{M}_s(s_t, R)$;
        $\tilde{d}_t = \mathcal{M}_e(d_t, R)$;
        **if** $f_j(s_t \oplus d_t, \tilde{s}_t \oplus \tilde{d}_t) > \mathcal{T}$ **then**
            Stop refining $s_t$ and $d_t$ in next round;
        $s_t = \tilde{s}_t$;
        $d_t = \tilde{d}_t$;

Figure 1: (a) Overview of Reflexion Framework. (b) Refinement via self-optimization algorithm.

35 from this result." We aim to detect these five types of errors to assist teachers in refining problem descriptions, ensuring they are more comprehensive and formal, which is essential for official examinations. Note that a single math world problem may contain multiple errors. Therefore, detecting error types in math word problems is a multilabel classification task.

## 5 PROMPT REFINEMENT THROUGH SELF-OPTIMIZATION

Various prompting methods have been proposed to enhance LLM performance. With zero-shot or few-shot learning (Brown et al., 2020), LLMs can tackle a range of tasks by providing a few examples about the task. Chain-of-thought prompting (CoT) techniques (Wei et al., 2022b; Yasunaga et al., 2024) successfully guide models to solve complex problems through step-by-step reasoning. Thus, we utilize LLMs with appropriate prompts for error type detection. Most research typically involves humans carefully designing prompts to instruct LLMs in performing tasks. We pose the question of whether allowing the model to independently understand the task objectives and reflect on its execution results to adjust its instructions and few-shot examples can enhance the performance of LLMs on this task. A mechanism may be needed to generate definitions that the model can understand and to provide suitable examples for specific error types that the model struggles with. We propose a method to iteratively reflect on incorrect classification results. By refining error type definitions based on these reflections, the model articulates task details according to its comprehension. Furthermore, the model selects proper examples to strengthen its reasoning abilities. Figure 1 (a) shows an overview of the framework for prompt refinement through self-optimization (PRO).

PRO comprises a classifier $\mathcal{M}_c$, a definition reflector $\mathcal{M}_s$, an example reflector $\mathcal{M}_e$, an evaluator $f_e$, and a judge $f_j$. $\mathcal{M}_c$, $\mathcal{M}_s$, and $\mathcal{M}_e$ are the same LLM but with different prompts. All the prompt templates used in this work are presented in Appendix B. For the self-optimization process, we partition our training set into sets $D$ and $D'$ for demonstration and reflection, respectively. In PRO, instead of using human-written definitions, we employ an LLM $\mathcal{M}$ to generate the initial definition $s_t$ of each error type $t \in T$. Specifically, given a set of questions $Q_t \in D$ that contains an error of $t$ and its corresponding corrections $Q'_t$, we obtain $s_t = \mathcal{M}(Q_t, Q'_t)$. To assess whether the definition requires refinement, we consult the classification results. Given the definitions $S$ of $T$, the selected examples $d$ in $D$, and the $i$-th question $q_i$ in $D'$, we obtain the prediction $y_i = \mathcal{M}_c(S, d, q_i)$. $\mathcal{M}_c$ is prompted to predict the possible error types in $q_i$.

After obtaining the initial classification results, we proceed to the self-optimization phase detailed in Figure 1 (b). We set a maximum number of trials to iteratively refine $S$ and update $d$ through $\mathcal{M}_s$ and $\mathcal{M}_e$, respectively. The evaluator $f_e$ measures the classification results using the macro-averaged F-score, allowing us to judge the model's performance in each round. For each $t$, incorrectly predicted questions are extracted for self-reflection. Formally, if the predicted error type $y_i^t$ does not match the ground truth $\hat{y}_i^t$, the question $q_i^t$ is appended to reflection set $R$. Subsequently, the refined definition $\tilde{s}_t$ of $t$ is obtained as $\tilde{s}_t = \mathcal{M}_s(s_t, R)$. Additionally, the demonstrations are updated by extracting questions in $R$. Given $R$ and the current demonstrations $d_t$ for $t$, we obtain the updated demonstrations $\tilde{d}_t = \mathcal{M}_d(d_t, R)$. $\tilde{s}_t$ and $\tilde{d}_t$ are used in the next round as $s_t$ and $d_t$, respectively.

Finally, to determine whether the updates to $s_t$ and $d_t$ have converged, judge $f_j$ compares the difference in the reflection results between the previous and current rounds. We measure the difference between the concatenation of $s_t \oplus d_t$ and $\tilde{s}_t \oplus \tilde{d}_t$ using ROUGE-1 (Lin, 2004). If the ROUGE-1 score $r = f_j(s_t \oplus d_t, \tilde{s}_t \oplus \tilde{d}_t)$ is greater than the threshold $\mathcal{T}$, we view it as convergence. In this case, $s_t$ and $d_t$ are not refined in the next round. If $S$ and $D$ converge, self-optimization ceases. To determine which definitions and examples to use, we measure the performance of $S$ and $d$ in each round using $z = f_e(\hat{y}, y)$: $S$ and $d$ from the highest-scoring round are used to instruct $\mathcal{M}_c$ to detect error types in the test set. The number of max trials and $\mathcal{T}$ are set to 10 and 0.9, respectively.

# 6 EXPERIMENTS

## 6.1 EXPERIMENTAL SETUP

Since most questions in Math23k were of the *None* type, we randomly selected a subset of math word problems from this type. This resulted in a total of 4,766 math word problems in MathError. To simulate limited data availability, we identified patterns from sparse data and applied them to specific tasks. Specifically, we focused on detecting error types in math word problems, despite having few such questions with errors. The data used for demonstration and reflection consisted of 15 and 30 math word problems, respectively. The remaining 4,721 math word problems were used for testing. The demonstration set consisted of 3 questions for each error type, providing a foundation for the initial model for inference. The reflection set, designed to refine and validate the model, included 2 questions for each error type and an additional 20 questions without errors. In the test set, the number of questions for *INTPN*, *Informal*, *Unit*, *Rel*, *Calc*, and *None* were 131, 1,071, 411, 601, 62, and 2,500, respectively.[3] We utilized OpenAI's API, in particular the `gpt-3.5-turbo-0125` and `gpt-4o` models. The temperature was set to 0 to increase reproducibility.

## 6.2 EXPERIMENTAL RESULTS

Table 3 shows the performance of each method on overall error types. We use the macro-averaged F-score as the evaluation metric and report the F-scores for each error type. The top four rows show the baseline models using LLM for direct inference on the test set with few-shot prompting and no refinement. The baseline models are GPT-3.5 and GPT-4o. We compare the impact of using the human-written and model-generated definitions. The "Definition" column denotes whether the initial definitions of all error types used in few-shot prompting were written by humans or generated by LLM. We also tested human-written and model-generated examples. In Table 3, all the results are based on human-written examples, as this is the optimal setting. The results of using human-written and model-generated examples will be discussed in the following section. Specifically, we employed an LLM to classify the error types of each question in the training set. Then, the LLM was instructed to generate examples based on the questions which are classified incorrectly.

As different reasoning strategies can yield different results, we conducted an experiment to find the most suitable strategy for detecting error types. We tested the following four strategies.

- Directly Classify: $\mathcal{M}_c$ is prompted to predict the error type with any reasoning steps.
- Analyze → Classify: $\mathcal{M}_c$ is tasked to generate an analysis of the errors present in the question and then classify the error types.

---

[3]The sum of questions exceeds the total because some contain multiple error types.

Table 3: Results of error type detection

| Method | Definition | Best strategy | Overall | INTPN | Informal | Unit | Rel | Calc | None |
|---|---|---|---|---|---|---|---|---|---|
| GPT-3.5 | Human | Solve → Classify | 0.2572 | 0.0519 | 0.3086 | 0.0616 | 0.3050 | 0.3394 | 0.4770 |
| | Model | Directly classify | 0.2295 | 0.0938 | 0.3422 | 0.1850 | 0.2681 | 0.2255 | 0.2625 |
| GPT-4o | Human | Solve → Classify | 0.2781 | 0.0777 | 0.4247 | 0.2319 | 0.2525 | 0.0461 | 0.6354 |
| | Model | Directly classify | 0.3118 | **0.1256** | **0.4362** | 0.3081 | 0.3930 | 0.2269 | 0.3812 |
| PRO (GPT-3.5) | Human | Solve → Classify | 0.2493 | 0.0134 | 0.2688 | 0.0702 | 0.2706 | 0.3270 | 0.5457 |
| | Model | Directly classify | 0.2547 | 0.0426 | 0.2063 | 0.1109 | 0.1393 | **0.3800** | **0.6490** |
| PRO (GPT-4o) | Human | Solve → Classify | 0.2697 | 0.0696 | 0.4108 | 0.2079 | 0.2747 | 0.0455 | 0.6096 |
| | Model | Directly classify | **0.3243** | 0.0936 | 0.4271 | **0.3109** | **0.4212** | 0.2538 | 0.4390 |

Table 4: Results of PRO (GPT-4o) using different inference strategies

| Method | Strategy | Overall | INTPN | Informal | Unit | Rel | Calc | None |
|---|---|---|---|---|---|---|---|---|
| PRO (GPT-4o) | CoT | 0.2367 | 0.0457 | 0.3368 | 0.2249 | 0.2097 | 0.0492 | 0.5534 |
| | Directly Classify | **0.3243** | 0.0936 | **0.4271** | **0.3109** | **0.4212** | **0.2538** | 0.4390 |
| | Analyze → Classify | 0.2563 | 0.0864 | 0.4106 | 0.2270 | 0.2878 | 0.0688 | 0.4574 |
| | Solve → Classify | 0.2420 | **0.0980** | 0.3942 | 0.1769 | 0.2934 | 0.1188 | 0.3708 |
| | Solve → Classify → Correct | 0.2709 | 0.0605 | 0.3187 | 0.1551 | 0.2898 | 0.1625 | **0.6385** |

- Solve → Classify: $\mathcal{M}_c$ solves the math word problem before classifying the error type. This strategy enables $\mathcal{M}_c$ to identify informal or missing information during problem-solving.
- Solve → Classify → Correct: after solving the problems and classifying the error types, $\mathcal{M}_c$ also corrects the errors in the questions.

For each method, we report the results of the best-performing strategy. In PRO, the self-optimization process refines human-written definitions and model-generated definition by $\mathcal{M}_s$. Experimental results show that PRO based on GPT-4o with model-generated definitions outperforms other methods, and significantly outperforms GPT-3.5 with model-generated definitions ($p < 0.05$). Although it does not significantly outperform the remaining methods in Table 3, overall, PRO (GPT-4o) with model-generated definitions detects the most problematic questions. This indicates that iteratively refining definitions and updating examples helps the model better accomplish the task.

Additionally, using model-generated definitions is more effective than using human-written ones, even though GPT-3.5 shows a different preference. This may be because in error type detection, reflecting on the wrong prediction and coming up with a solution to refine the definitions and select examples that can strengthen the reasoning process requires an LLM with more powerful natural language understanding and generation abilities. Comparing the results between GPT-4o and "PRO (GPT-4o)", the proposed reflection mechanism enhances the ability to detect *Informal*, *Rel*, *Calc*, and *None* types. Although the F-score for *None* type detection is not the highest, we may require a more stringent model in this context, even if it entails accepting false alarms. This supports our hypothesis that allowing the model to generate its own decision criteria leads to better task comprehension and enhances the reasoning ability.

In contrast, we find that GPT-4o does poorly at detecting the *Calc* type, especially when using human-written definitions. Perhaps LLMs possess better language understanding capability, allowing them to conjecture the most possible calculation order described in natural language. Hence, the model perceives the descriptions as without errors. Moreover, both GPT-3.5 and GPT-4o struggle to detect *INTPN*. This shows that detecting whether a math word problem has multiple interpretations is still a challenging issue. Further error analysis is presented in Appendix C.

Table 4 shows the results of "PRO (GPT-4o)" using different strategies. In general, direct classification is the most effective for detecting problematic questions, whereas solving the problem and then classifying its error type is best for detecting *INTPN* errors. Interestingly, if the strategy involves the LLM correcting the problematic questions, the model tends to classify questions as correct. We also conducted a comparison using the CoT prompting method. Unexpectedly, this approach yielded poorer performance compared to the "Direct Classify" method. Based on these results, we specu-

Table 5: Comparison of GPT-3.5 with different prompting methods for each error type

| Method | Strategy | Overall | INTPN | Informal | Unitless | Rel | Calc | None |
|---|---|---|---|---|---|---|---|---|
| GPT-3.5 Zero-Shot | | 0.2202 | 0.0262 | 0.1620 | **0.2862** | **0.3238** | 0.2704 | 0.2524 |
| GPT-3.5 Few-Shot | Directly Classify | **0.2407** | **0.0875** | **0.2729** | 0.2014 | 0.2457 | 0.2363 | 0.4001 |
| GPT-3.5 CoT | | 0.2196 | 0.0000 | 0.1667 | 0.2382 | 0.0458 | **0.2884** | **0.5784** |

Table 6: Results of different classification methods with zero-shot prompting

| Method | Definition | Task | Strategy | Overall | INTPN | Informal | Unit | Rel | Calc | None |
|---|---|---|---|---|---|---|---|---|---|---|
| | Human | Binary | | 0.1447 | **0.0541** | **0.3674** | 0.1620 | 0.2586 | 0.0259 | 0.0000 |
| GPT-3.5 Zero-shot | Human | Multilabel | Directly Classify | 0.2202 | 0.0262 | 0.1620 | 0.2862 | **0.3238** | **0.2704** | 0.2524 |
| | Model | Multilabel | | **0.2494** | 0.0309 | 0.1442 | **0.3864** | 0.2826 | 0.2607 | **0.3916** |

late that if an LLM is guided to reason, the stronger the LLM's inferencing ability, the more it may overlook issues in textual descriptions, leading to ineffective error detection.

# 7 ANALYSIS AND DISCUSSION

In this section, we formulate and discuss six research questions which we address by conducting corresponding experiments. The results reported below are based on the "directly classify" strategy for each method. This pilot study seeks to explore how in-context learning can be utilized to detect problematic questions. Hence, the first research question (**RQ1**) arises: Which of the commonly used prompting methods—zero-shot, few-shot, or CoT prompting—yields the best performance in detecting errors in math word problems?

**Impact of Different Prompting Methods:** Table 5 presents the results of utilizing GPT-3.5 to detect error types using different prompting methods. The results indicate that few-shot prompting based on human-written definitions and examples outperforms the other prompting methods, although the difference is not statistically significant. We found that CoT prompting struggles with detecting questions that have multiple interpretations. The analysis of the model's reasoning output indicates that CoT prompting often leads the model to infer a fixed interpretation of the question, overlooking alternative interpretations. In contrast, GPT-3.5 with few-shot prompting demonstrates promising performance in detecting questions with multiple interpretations, informal wording, and unclear relationships. These error types are critical for teachers when refining problem design, as they significantly influence the clarity and validity of the questions posed to students.

In the experiments in Table 3, we set error type detection as multilabel classification. However, the model can also perform binary classification for each error type by querying whether a specific math word problem contains that particular error. This raises the second research question (**RQ2**): Which method for error type detection is more suitable: multilabel classification or binary classification?

**Multilabel Classification vs. Binary Classification:** Table 6 shows the results when treating error type detection as multilabel or binary classification. We used the GPT-3.5 model with zero-shot prompting. The strategies in all settings are "Directly classify". We find that multilabel classification outperforms binary classification. Additionally, binary classification causes the model to tend to view every math word problem as erroneous. A possible reason is that the prompt for multi-label classification provides definitions of all types, which helps the model better understand the error types compared to binary classification. However, as shown in Table 6, the performance when using the GPT-3.5 definitions is better than that using the human-written definitions. This leads to the third research question (**RQ3**): Should the definitions and examples in this task be generated by the model itself or provided by humans?

**Human-Written vs. Model-Generated Prompts:** Table 7 presents the results when using GPT-3.5 with few-shot prompting with various settings to confirm which definitions and examples are better. The overall best performance is achieved when human definitions are written and paired with human-written examples. However, as shown in Table 7, the average performance in detecting five error types using GPT-3.5-generated definitions is better than that using human-written definitions.

Table 7: Model-generated vs. human-written definitions and examples

| Method | Definition | Examples | Strategy | Overall | INTPN | Informal | Unit | Rel | Calc | None |
|---|---|---|---|---|---|---|---|---|---|---|
| GPT-3.5 Few-shot | Human | Human | Directly Classify | **0.2407** | 0.0875 | 0.2729 | 0.2014 | 0.2457 | 0.2363 | **0.4001** |
| | Human | Model | | 0.2031 | 0.0596 | 0.2004 | 0.2030 | 0.2510 | **0.2701** | 0.2345 |
| | Model | Human | | 0.2295 | **0.0938** | **0.3422** | 0.1850 | **0.2681** | 0.2255 | 0.2625 |
| | Model | Model | | 0.2267 | 0.0541 | 0.2665 | **0.2471** | 0.2307 | 0.2218 | 0.3400 |

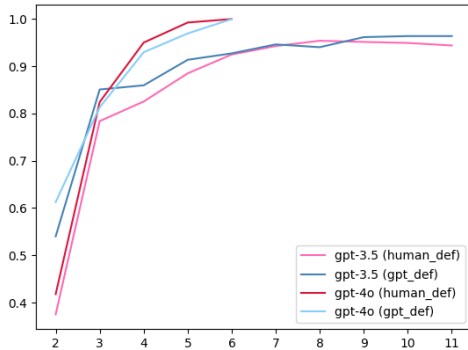

Figure 2: Refinement differences across rounds using PRO

The average F-scores of all the error types in the first and third rows are 0.2088 and 0.2229, respectively. Comparing Table 6 and Table 7, "GPT-3.5 Zero-shot" using model-generated definitions (0.2494) outperforms "GPT-3.5 Few-shot" using human-written definitions and examples (0.2407). Nonetheless, the few-shot prompting methods are better at detecting errors in the "INTPN" and "Informal" types. Thus, when seeking to detect error types, a combination of model-generated definitions and human-written examples is a more suitable approach. On the other hand, although PRO achieves better performance by refining prompts based on reflecting on wrong predictions, the changes in prompts for each round have not been discussed. This leads to the forth research question (**RQ4**): How do different initial definition approaches and LLMs impact the refinement differences and convergence across rounds?

**Effects on Refinement and Convergence:** Figure 2 presents the refinement differences across rounds using PRO. We present the results of PRO based on GPT-3.5 and GPT-4o, using either human-written or model-generated definitions, with all examples human-written. The x-axis represents the round number, and the y-axis denotes the ROUGE score difference between the current and the previous round's definitions and examples. Specifically, we measured the ROUGE-1 score of definitions and examples before and after rounds of self-optimization. A higher score indicates greater similarity between definitions and examples across rounds. With this analysis we investigate the number of rounds required for convergence under different settings. GPT-3.5 significantly refines descriptions, leading to considerable variation and making convergence difficult, often reaching the maximum of 10 rounds. GPT-4o, in turn, makes slight refinements, typically adding a few details to definitions and examples, and generally converging in around 5 rounds due to smaller refinements and fewer classification errors.

Additionally, Figure 2 shows a significant increase in ROUGE scores during the first three rounds, indicating substantial modifications to definitions and examples based on classification results. This trend levels off in later rounds, suggesting the model has exhausted useful information from $D^{'}$ for further refinement. Moreover, with human-written definitions, the ROUGE score increases more sharply in the first three rounds compared to model-generated definitions, indicating a greater initial disparity. Examples of refinement results are in Appendix E.

In the above experiments, we evaluated only GPT-3.5 and GPT-4o, two large LLMs. This leads to the fifth research question (**RQ5**): How do LLMs of different sizes perform on this task?

**Performance of LLMs with Different Parameter Sizes**: We compared the performance of different sizes of LLaMA 3 (AI@Meta, 2024) with few-shot prompting using model-generated definitions

Table 8: Results using different LLMs

| Method | Denifition | Examples | Strategy | Overall | INTPN | Informal | Unit | Rel | Calc | None |
|---|---|---|---|---|---|---|---|---|---|---|
| LLaMA3 8B | | | | 0.1479 | 0.0308 | 0.0165 | 0.2272 | 0.2804 | 0.1362 | 0.1963 |
| LLaMA3 70B (8 bit) | | | Directly | 0.2031 | 0.0181 | 0.2406 | 0.2748 | 0.3670 | 0.2253 | 0.0926 |
| LLaMA3 70B | Model | Model | Classify | 0.2210 | 0.0138 | **0.2488** | 0.2853 | **0.4066** | 0.2443 | 0.1271 |
| GPT-3.5 175B | | | | **0.2494** | **0.0309** | 0.1442 | **0.3864** | 0.2826 | **0.2607** | **0.3916** |

and examples. The results are shown in Table 8. GPT-3.5 outperforms the other models in overall performance. Although LLaMA3 70B excels at detecting errors, it struggles to identify error-free questions. It tends to consider questions as informal or incomplete. As model size decreases, overall performance also decreases. Therefore, effective detection of incorrect descriptions in math word problems requires a large LLM with strong semantic understanding capabilities.

In our previous research questions, we find that model-generated definitions are most suitable for detecting error types. We further investigate the reasons behind this finding with the sixth research question (**RQ6**): Why are model-generated definitions more effective?

**Impact of Model-Generated and Human-Written Prompts**: Based on the results shown in Table 8, we compared the perplexity of human-written prompts and model-generated prompts. We speculate that model-generated prompts better align with the model's probability distribution, enhancing its semantic understanding and ability to detect error types. To this end, we examine the performance of LLaMA3 8B[4] using human-written and model-generated definitions with examples, with overall performances of 0.1191 and 0.1479, respectively. The results are shown in Table 18 in Appendix D. These results align with the trends presented in Table 6. Furthermore, we assess the model's perplexity on both human-written and model-generated definitions with examples, resulting in values of 0.7041 and 0.6499, respectively. These findings suggest that model-generated prompts are more predictable, implying the model handles this type of narrative better. This may explain why model-generated definitions are more suitable in this context.

## 8 CONCLUSION

Recent years have witnessed a surge of work on ambiguous and unanswerable questions. In contrast to previous studies, this paper focuses on detecting issues in math word problems that can lead to multiple solutions or render them unanswerable. These issues arise from different interpretations and misunderstandings due to imprecise problem descriptions or missing information needed for problem-solving. We present a task on math word problem design support and construct MathError, the first human-annotated dataset, to explore the method of error type detection. We explore prompt refinement through self-optimization (PRO) to instruct an LLM to adapt to the given task. We investigate whether definitions for error types and corresponding few-shot examples are more effectively provided by humans or models. The results show that machine-generated definitions of error types, supplemented by human-written examples, enhance effectiveness in error type detection. During the self-optimization process, we modify the definitions based on the classification errors of the data retained from the training set, and allow the LLM to select which examples to add. However, accurately identifying the errors in a math word problem is still challenging; more advanced methods are left as future work. We also plan to explore methods for correcting problematic questions. At the current stage, this work has several limitations. We extend the Math23K dataset with error type annotation. While this is the first human-annotated dataset used for detecting errors in math word problems, our research was limited to Chinese math word problems and did not explore other languages. Additionally, our work currently relies on a single dataset comprising questions collected from an online educational platform, predominantly at the elementary school level, which may not offer sufficient diversity in problem types. Moreover, we define only five error types within this dataset: these may not encompass all possible errors in math word problems. At this stage, our primary goal is to provide teachers with identified error types in math word problems, though considering factors like students' prior knowledge and cognitive barriers remains important for future research.

---

[4]Because calculating probability distributions through the API is difficult, we use an LLM that runs locally.

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

## A  ADDITIONAL ANNOTATION DETAILS

Table 9 presents the annotation guidelines provided to the annotators. We invited native Chinese speakers as annotators, all of whom were from universities in the same country as the authors. Before beginning the annotation, we explained the purpose of the data and confirmed the compensation terms with the annotators. Compensation was calculated based on the minimum wage regulations of the authors' country; both parties agreed to these terms.

Table 9: Annotation guidelines

| |
|---|
| **Objective:** |
| To create a dataset of Chinese math word problems with error type annotations, where some problems contain errors that need to be annotated accordingly. The error types include misleading, unclear problem statement, etc. |
| **Reminders:** |
| - We will regularly track your annotated results, so any questions can be addressed immediately. |
| - If you find an error with a problem after reviewing it, select the appropriate category from the list below and label it. |
| - Label the chosen error type with a lowercase "x" in the box following the problem. |
| - Multiple annotations can be applied to a single problem. |
| **Error Type Definition:** |
| *Described in Section 3* |

Table 10: Prompt template for event type detection (directly classify method).

**Task Introduction:**
Please analyze and identify whether the [given problem] contains any of the following errors based on the descriptions under [error types]: multiple interpretations, informal wording, unitless, unclear relationship, or calculation error. If any errors are identified, respond with the corresponding option(s). If the [Given Problem] does not contain any of these errors, select (F). The problem must match the definitions of the error types to be considered erroneous; note that a single problem may contain multiple errors.
[Error Types]
(A) Multiple Interpretations: {*Definition*} {*Examples*}.
(B) Informal Wording: {*Definition*} {*Examples*}.
(C) Unitless: {*Definition*} {*Examples*}.
(D) Unclear Relationship: {*Definition*} {*Examples*}.
(E) Calculation Error: {*Definition*} {*Examples*}.
(F) None of the above. Example: {*Example*}

**Response Requirement:**
Enter your predicted results under [classification result]. If multiple errors are present, separate them with a semicolon ";".
Example response:
[Given Problem] There are 120 chicken eggs, and the duck eggs are 1/6 more than the chicken eggs. How many eggs are there in total?
[Classification Result] (C) Informal Wording; (E) Unclear Relationship
[Given Problem] {*question*}

Table 11: Prompt template for event type detection (Analyze→Classify method).

**Task Introduction:**
{Same as that shown in Table 10}

**Response Requirement:**
Please provide an analysis of up to 150 words after [Analysis] and record your assessment results under [Classification Result]. If multiple errors are present, separate them with a semicolon ";".
Example response:
[Given Problem] There are 120 chicken eggs, and the duck eggs are 1/6 more than the chicken eggs. How many eggs are there in total?
[Analysis] The statement is unclear, rendering the meaning of "1/6" ambiguous. This could imply that the number of duck eggs is 1/6 that of the chicken eggs (which is logically inconsistent as it contradicts the statement that there are more duck eggs), or that the duck eggs exceed the number of chicken eggs by 1/6 of the chicken egg count.
[Classification Result] (C) Informal Wording; (E) Unclear Relationship
[Given Problem] {*question*}

# B  INPUT FORMATS

Tables 10, 11, 12, and 13 contain templates of the "Directly classify", "Analyze → Classify", "Solve → Classify", and "Solve → Classify → Correct" strategies, respectively. The templates for definition and example generation are shown in Tables 14 and 15. The proposed definition and example refinement templates are shown in Tables 16 and 17. In the refinement prompts, we instruct the LLM to analyze why the error was not detected. However, we find that not including the result of the analysis is better. This may be because the LLM still struggles to complete the error type detection tasks. The content in the generated analysis may contain incorrect information.

# C  ERROR ANALYSIS

Figure 3 presents the confusion matrix for error type detection results using the PRO (GPT-4o) method with model-generated definitions and human-written examples.[5] *Multiple Interpretations*

---

[5]The total numbers in the confusion matrix differ from the actual number of error types because our task is multilabel classification. An error type can be incorrectly identified as multiple different error types.

Table 12: Prompt template for event type detection (Solve → Classify method).

**Task Introduction:**
{Same as that shown in Table 10}

**Response Requirement:**
Please solve the problem and generate your calculations and reasoning within 150 words under `[Calculation Process and Rationale]`. If the solution remains elusive, it is possible that the problem includes the above errors. Based on the difficulties encountered during the calculation, place your predicted results behind `[Classification Result]`. If multiple errors are identified, separate them with a semicolon ";".
`Example response:`
`[Given Problem]` There are 120 chicken eggs, and the duck eggs are 1/6 more than the chicken eggs. How many eggs are there in total?
`[Calculation Process and Rationale]` The statement "duck eggs are 1/6 more" is ambiguous: does it refer to 1/6 more than the number of duck eggs, or 1/6 more than the number of chicken eggs? The lack of a clear referent for the ratio leads to confusion. If it means 1/6 more than the number of duck eggs, the question lacks sufficient information to calculate the number of duck eggs. Conversely, if it refers to 1/6 of the chicken eggs, then the number of duck eggs would be 120 + 120/6 = 140.
`[Classification Result]` (C) Informal Wording; (E) Unclear Relationship
`[Given Problem]` {*question*}

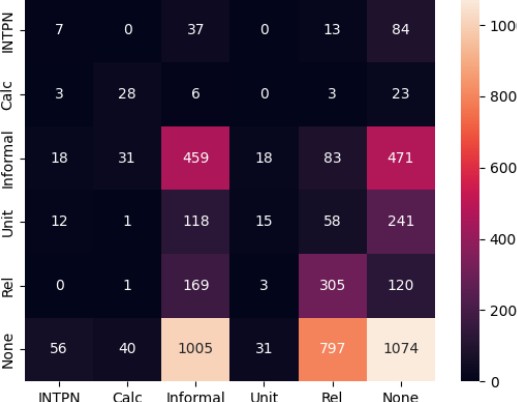

Figure 3: Confusion matrix for error type detection using "PRO (GPT-4o)" method with model-generated definitions and human-written examples

and *Unitless* errors are often predicted as *None* or *Information Wording*. Additionally, questions exhibiting the *Unclear Relationship* error are frequently predicted as *Informal Wording*.

Interestingly, math word problems with no errors (belongs to *None*) are often predicted as *Informal Wording* and *Unclear Relationship*. To understand this behavior, we examined the results predicted by other methods, which similarly tended to predict *None* type questions as containing *Informal Wording* and *Unclear Relationship* errors. This may be due to using *Informal Wording* and *Unclear Relationship* errors as examples to demonstrate the response format.

To investigate the impact of the response format demonstration, we replaced the response format demonstration with *Calculation Error*. Figure 4 shows the confusion matrix of error type detection results after changing the format demonstration. The model's preference for predicting *Informal Wording* and *Unclear Relationship* errors became less pronounced, but there was no tendency to predict the *None* type as *Calculation Error*. From this result, we find that the model is highly sensitive to the examples used. However, it remains unclear why the prediction trend does not align with our original hypothesis that the model will predict the error types used in the format demonstration after changing the example.

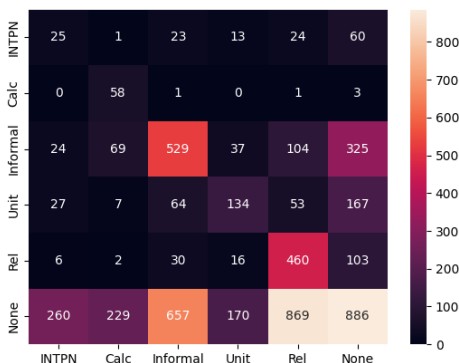

Figure 4: Confusion matrix for error type detection with format example changed to *Calc*

Table 13: Prompt template for event type detection (Solve→Classify→Correct method).

**Task Introduction:**

{Same as that in Table 10}

**Response Requirement:**

First, please solve the problem and document your calculation process and reasoning within 150 words under `[Calculation Process and Rationale]`. If you are unable to determine an answer, the problem likely contains the errors described above. Based on any difficulties encountered during the calculation, record possible error types under `[Classification Result]`. If multiple errors are identified, separate them with a semicolon ";". Afterwards, try to modify the problem and note the revised version under `[Corrected Problem]`. ";".

`Example response:`

`[Given Problem]` There are 120 chicken eggs, and the duck eggs are 1/6 more than the chicken eggs. How many eggs are there in total?

`[Calculation Process and Rationale]` The statement "duck eggs are 1/6 more" is ambiguous: does it refer to 1/6 more than the number of duck eggs, or 1/6 more than the number of chicken eggs? The lack of a clear referent for the ratio leads to confusion. If it means 1/6 more than the number of duck eggs, the question lacks sufficient information to calculate the number of duck eggs. Conversely, if it refers to 1/6 of the chicken eggs, then the number of duck eggs would be 120 + 120/6 = 140.

`[Classification Result]` (C) Informal Wording; (E) Unclear Relationship

`[Corrected Problem]` There are 120 chicken eggs, and the duck eggs are 1/6 more than the number of chicken eggs. How many eggs are there in total?

`[Given Problem]` {*question*}

## D COMPARISON OF ERROR TYPE DETECTION RESULTS BETWEEN HUMAN-WRITTEN DEFINITIONS AND MODEL-GENERATED PROMPTS

Table 18 reports the comparison between using human-written and model-generated prompts under LLaMA 3 8B.

## E EXAMPLES OF PROMPT REFINEMENT RESULTS

Tables 19, 20, 21, 22, and 23 show the examples of prompt refinement results for *INTPN*, *Informal*, *Unit*, *Rel*, and *Calc*, respectively. The following prompt refinement results in our study are presented in Chinese. To facilitate reader understanding, we used GPT-4o to translate them into English.

Table 14: Prompt template for generating definitions by model.

**Task Introduction:**
You are tasked with writing a definition for the error type based on problematic questions and their corrected versions. Please use the following `[Problematic Question]` and corresponding `[Corrected Question]` to formulate your definitions. Please place the generated definition behind `[Definition]`.

`[Given Error Type]` {*error type*}
`[Problematic Question 1]` {*problematic question* 1}
`[Corrected Question 1]` {*corrected question* 1}
`[Problematic Question 2]` {*problematic question* 2}
`[Corrected Question 2]` {*corrected question* 2}
...
`[Definition]`

Table 15: Prompt template for generating examples by model.

**Task Introduction:**
You are tasked to provide both positive and negative examples for each error category to enhance the precision of our classification system. Based on the definitions of error types and the classification results of `[Given Question]`, please supply one example that fits each error category and one that does not. Be mindful that discrepancies between `[Classification Result]` and `[Correct Classification]` indicate wrong prediction; analyze the reasons of these issues to generate more challenging examples and place them behind `[Examples]`.

`[Error Types]`
(A) Multiple Interpretations: {*Definition*} {*Examples*}.
(B) Informal Wording: {*Definition*} {*Examples*}.
(C) Unitless: {*Definition*} {*Examples*}.
(D) Unclear Relationship: {*Definition*} {*Examples*}.
(E) Calculation Error: {*Definition*} {*Examples*}.
(F) None of the above. Example: {*Example*}

`[Given Question 1]` {*Question* 1}
`[Classification Result 1]` {*Result* 1}
`[Correct Classification]` 1 {*Annotated error type* 1}

`[Given Question 2]`{*Question* 2}
`[Classification Result 2]`{*Result* 2}
`[Correct Classification]` 2 {*Annotated error type* 2}

...
`[Examples]`

Table 16: Prompt template for definition refinement.

**Task Introduction:**
You are tasked with refining the given error type definition based on the results of erroneous classifications. The classification task involves analyzing the following `[Error Types]` to determine whether `[Given Problem]` contain errors such as multiple interpretations, informal wording, unitless, unclear relationship, or calculation error. If any errors are present, select the corresponding option. If the `[Given Problem]` contains no such errors, choose (F). The problem must match the definitions of the error types to be considered erroneous, and note that a single problem may contain multiple errors.
`[Error Types]`
(A) Multiple Interpretations: {*Definition*} {*Examples*}.
(B) Informal Wording: {*Definition*} {*Examples*}.
(C) Unitless: {*Definition*} {*Examples*}.
(D) Unclear Relationship: {*Definition*} {*Examples*}.
(E) Calculation Error: {*Definition*} {*Examples*}.
(F) None of the above. Example: {*Example*}

{*wrong predictions*}
Please refer to both the correctly classified results and the misclassified ones under "{target type}" error, along with the definition of this error type. Analyze why there was a wrong prediction or why the error was not recognized. Use this analysis to refine the definition of the `[Error Type]` "{target type}".
Please output the refined definition in the following format: `[Error Type Definition]`: [Revised Definition] [Analysis]

Table 17: Prompt template for example refinement.

**Task Introduction:**
{Same as that in Table 16}

{*wrong predictions*}
Please refer to both the correctly classified results and the misclassified ones under "{target type}" error, along with the definition of this error type. Analyze why there was a wrong prediction or why the error was not recognized. Use this analysis to refine the examples of the `[Error Type]` "{target type}".
`[Error Type]`: {*target type*}
Please output the refined examples in the following format: `[Error Type Example]`: [Refined Example] [Analysis]

Table 18: Effect of Human-Written vs. Model-Generated Prompts on LLaMA 8B Performance

| Method | Definition | Overall | INTPN | Informal | Unit | Rel | Calc | None |
|---|---|---|---|---|---|---|---|---|
| Llama 8B | Human | 0.1191 | 0.0189 | 0.0037 | **0.2399** | 0.1685 | **0.2084** | 0.0751 |
| | Model | **0.1479** | **0.0308** | **0.0165** | 0.2272 | **0.2804** | 0.1362 | **0.1963** |

Table 19: Example of prompt refinement for the multiple-interpretation type

| Error Type | Initial Definition and Example Generated by LLM | Refined By LLM in the 2nd Round |
|---|---|---|
| Multiple Interpretations | 多解是指在数学题目中，由于题目描述不够明确或存在歧义，导致读者可以从不同的角度理解题意，从而得出多个不同的解答。这种情况下，题目本身并没有唯一的解答，或者解答的条件不够充分，无法确定唯一的正确答案。范例A. 学校购买每张单价是140元的课桌，买了30张还多480元．如果用这笔钱买椅子，可以买40把．每把椅子的单价=多少元? B. 一块平行四边形的街头广告牌，底是12.5米，高是6.4米．如果要油饰这块广告牌，每平方米用油漆0.6千克，需要多少千克油漆? 其中A 因为题目未提到这笔钱刚好只能买这些椅子，所以答案有无限多种可能。B 未提到油漆要涂双面还是单面，属于此类。(Multiple interpretations in problem-solving occurs when the description of a mathematical problem is unclear or contains ambiguities, allowing readers to interpret the problem from different perspectives and arrive at multiple distinct solutions. In such cases, the problem itself does not have a unique solution, or the conditions provided are insufficient to determine a single correct answer.For example:A. A school purchases desks at a unit price of 140 units, buying 30 desks with an additional 480 units remaining. If this remaining amount is used to purchase chairs, they can buy 40 chairs. What is the unit price of each chair? B. A street advertisement board in the shape of a parallelogram has a base of 12.5 meters and a height of 6.4 meters. If painting this board requires 0.6 kilograms of paint per square meter, how many kilograms of paint are needed? In the case A, since the problem does not specify that the remaining money can only be used to buy these chairs, there are infinitely many possible answers.The problem of case B falls into the error type of multiple interpretations since it does not specify whether the paint is needed for both sides or just one side.) | 多解是指在数学题目中，由于题目描述不够明确或存在歧义，导致读者可以从不同的角度理解题意，从而得出多个不同的解答。这种情况下，题目本身并没有唯一的解答，或者解答的条件不够充分，无法确定唯一的正确答案。具体来说，这可能包括但不限于以下情况：题目中缺少关键信息或条件，导致无法确定唯一解；题目中存在多种可能的解释或理解方式，导致多个解答；题目中使用了模糊或不明确的语言，导致解答者对题意产生不同的理解。范例A. 学校购买每张单价是140元的课桌，买了30张还多480元．如果用这笔钱买椅子，可以买40把．每把椅子的单价=多少元? B. 一块平行四边形的街头广告牌，底是12.5米，高是6.4米．如果要油饰这块广告牌，每平方米用油漆0.6千克，需要多少千克油漆? C. 小芳一家准备出去旅游，计划每天的支出如下：住宿费65元，交通费25元，伙食费60元，如果外出旅游一周（按7天算），至少应准备多少钱? 其中A 因为题目未提到这笔钱刚好只能买这些椅子，所以答案有无限多种可能。B 未提到油漆要涂双面还是单面，属于此类。C 属于此类。(Multiple interpretations refer to situations in mathematical problems where the problem description is not clear enough or contains ambiguities, leading readers to interpret the problem from different perspectives and arrive at multiple different solutions. In such cases, the problem itself does not have a unique solution, or the conditions provided are insufficient to determine a single correct answer. Specifically, this may include, but is not limited to, the following situations: the problem lacks key information or conditions, making it impossible to determine a unique solution; the problem allows for multiple interpretations or understandings, leading to multiple interpretations; the problem uses vague or unclear language, resulting in different understandings by the solvers.Examples:A. A school purchases desks at a unit price of 140 units, buying 30 desks with an additional 480 units remaining. If this remaining amount is used to purchase chairs, they can buy 40 chairs. What is the unit price of each chair?B. A street advertisement board in the shape of a parallelogram has a base of 12.5 meters and a height of 6.4 meters. If painting this board requires 0.6 kilograms of paint per square meter, how many kilograms of paint are needed?C. Xiaofang's family plans to go on a trip, with the following daily expenses: accommodation 65 units, transportation 25 units, and food 60 units. If they plan to travel for a week (7 days), how much money should they prepare at least?In example A, the problem does not specify that the remaining money can only be used to buy these chairs, so there are infinitely many possible answers. In example B, it is unclear whether the paint is needed for both sides or just one side, making it fall into this error type. Example C also falls into this error type.) |

Table 20: Example of prompt refinement for the informal wording type

| Error Type | Initial Definition and Example Generated by LLM | Definition Refined By LLM in the 2nd Round |
|---|---|---|
| Informal Wording | 当数学题目中缺少关键的叙述或符号，导致题意不明确或无法正确理解时，这种情况被称为「不明或缺失关键叙述」。这种错误会使读者无法准确地解读题目，从而影响解题过程和结果。
范例A. 一本书原__是18元，降价一半后售价是多少？ B. 一个数字的1/3是另一个数字的1/4，这两个数字的比是((())/( ()))其中，A 缺少部分叙述，让题目略显怪异，属于此类，正确的写法：一本书原价是18元。而B 的括号为答案区，并非缺少资讯，因此不在此列。
(When a mathematical problem lacks essential descriptions or symbols, resulting in unclear or incorrect interpretation. Such errors prevent readers from accurately understanding the problem, thereby affecting the problem-solving process and outcomes.
For example:
A. The original price of a book was __18 units. What is the price after a 50% discount?
B. One number is 1/3 of another number's 1/4. The ratio of these two numbers is ((())/(())).
In example A, the lack of critical descriptions makes the problem somewhat peculiar, falling into this error type. The correct statement should be: "The original price of a book is 18 units." In example B, the parentheses indicate the answer area and do not represent missing information; therefore, it is not considered under this error type.) | 当数学题目中缺少关键的叙述或符号，导致题意不明确或无法正确理解时，这种情况被称为「不明或缺失关键叙述」。这种错误会使读者无法准确地解读题目，从而影响解题过程和结果。常见的情况包括：题目中缺少必要的数据或条件，导致无法进行计算；题目叙述不完整或存在歧义，使得解题者无法确定应该如何解答。
范例A. 一本书原__是18元，降价一半后售价是多少？ B. 一个数字的1/3是另一个数字的1/4，这两个数字的比是((())/( ()))C. 四年级共有13个合作小组，每个小组有4名同学．已知全班同学本学期共得到312颗合作星，平均每个同学得到几颗合作星？D. 6（2）班有科技书240本，故事书比科技书少(1/6)．故事书比科技书少多少书？其中，A 缺少部分叙述，让题目略显怪异，属于此类，正确的写法：一本书原价是18元。而B 的括号为答案区，并非缺少资讯，因此不在此列。C 属于此类，D 属于此类。
(When a mathematical problem lacks critical descriptions or symbols, making the problem statement unclear or difficult to interpret correctly, this situation is referred to as "Informal Wording" Such errors prevent readers from accurately interpreting the problem, thus affecting the problem-solving process and outcomes. Common issues include missing necessary data or conditions that make calculation impossible, or incomplete or ambiguous problem statements that leave solvers uncertain about how to approach the problem.
Examples:
A. The original price of a book was __18 units. What is the price after a 50% discount?
B. One number is 1/3 of another number's 1/4. The ratio of these two numbers is ((())/(())).
C. There are 13 cooperative groups in the fourth grade, each with 4 students. If the entire class earned a total of 312 cooperation stars this semester, how many stars did each student earn on average?
D. Class 6(2) has 240 science books. The number of storybooks is less by (1/6). How many fewer storybooks are there compared to science books?
In example A, the lack of a part of the statement makes the problem somewhat peculiar and falls into this error type. The correct statement should be: "The original price of a book is 18 units." Example B uses parentheses to denote the answer area and does not lack information; therefore, it is not considered under this error type. Examples C and D fall into this error type due to missing or unclear descriptions.) |

Table 21: Example of prompt refinement for the unit-less type

| Error Type | Initial Definition and Example Generated by LLM | Definition Refined By LLM in the 2nd Round |
|---|---|---|
| Unit-less | 缺少单位'是一种常见的数学错误，指的是在数学题目或答案中缺少必要的单位标示，导致读者无法准确理解数值的具体含义。单位是用来表示数量的标准，如千米、米、千米/时、元等。在数学问题中，单位的缺失会使得问题或答案不完整，甚至可能引起误解或错误计算。
范例A. 长方体水槽长3公尺、宽2公尺、高5公尺，请问水槽的容量是? B. 工人一天可以赚10元，工作五天可以赚多少? C. 110-10*6,其中，A 会有其他可能性(立方公分,立方公尺) 属于此类错误，正确的写法: 立方公尺。而B，钱的单位默认是'元'，所以不算在此列。C 为单纯的数学题目，因此不在此列。
("Unit-less" is a common mathematical error, referring to the absence of necessary unit indicators in a problem or its solution, which prevents the reader from accurately understanding the specific meaning of the values. Units are standards used to express quantities, such as kilometers, meters, kilometers per hour, currency units, etc. In mathematical problems, the absence of units can render the problem or solution incomplete, potentially leading to misunderstandings or calculation errors.
Examples:
A. A rectangular water tank has a length of 3, a width of 2, and a height of 5. What is the capacity of the tank?
B. A worker can earn 10 per day. How much can they earn in five days?
C. Calculate 110 - 10 * 6.
In example A, the lack of units (e.g., cubic centimeters, cubic meters) makes it fall into this error type of error. The correct specification should be cubic meters. In example B, the unit of money is implicitly understood as "currency units," so it is not considered under this error type. Example C is purely a mathematical expression and does not fall under this error type either.) | "缺少单位"是一种常见的数学错误，指的是在数学题目或答案中缺少必要的单位标示，导致读者无法准确理解数值的具体含义。单位是用来表示数量的标准，如千米、米、千米/时、元等。在数学问题中，单位的缺失会使得问题或答案不完整，甚至可能引起误解或错误计算。特别是当题目涉及多个不同的数值或比例时，缺少单位会使得这些数值之间的关系不明确，从而影响解题的准确性。
范例A. 长方体水槽长3公尺、宽2公尺、高5公尺，请问水槽的容量是? B. 工人一天可以赚10元，工作五天可以赚多少? C. 110-10*6 D. 修路队修一条公路，第一天修了全长的(1/4)，第二天修了全长的20%，两天共修了3.6千米，这条公路有多长? 其中，A 会有其他可能性(立方公分,立方公尺) 属于此类错误，正确的写法: 立方公尺。而B，钱的单位默认是'元'，所以不算在此列。C 为单纯的数学题目，因此不在此列。D 属于此类。
(Unit-less is a common mathematical error, referring to the absence of necessary unit indicators in a problem or its solution, which prevents readers from accurately understanding the specific meaning of the values. Units are standards used to express quantities, such as kilometers, meters, kilometers per hour, currency units, etc. In mathematical problems, the absence of units can render the problem or solution incomplete, potentially leading to misunderstandings or calculation errors. Particularly when problems involve multiple different values or ratios, missing units can obscure the relationships between these values, thus affecting the accuracy of the solution.
Examples:
A. A rectangular water tank is 3 long, 2 wide, and 5 high. What is the tank's capacity?
B. A worker can earn 10 per day. How much can they earn in five days?
C. Calculate 110 - 10 * 6.
D. A road construction team repairs a road. On the first day, they repaired (1/4) of the total length, and on the second day, they repaired 20% of the total length. In two days, they repaired 3.6 kilometers. How long is the road?
In example A, the lack of units (e.g., cubic centimeters, cubic meters) makes it fall into this error type of error. The correct specification should be cubic meters. In example B, the unit of money is implicitly understood as "currency units," so it is not considered under this error type. Example C is a purely mathematical problem and does not fall into this error type. Example D lacks units for the road length and falls into this error type.) |

Table 22: Example of prompt refinement for the unclear relationship type

| Error Type | Initial Definition and Example Generated by LLM | Definition Refined By LLM in the 2nd Round |
|---|---|---|
| Unclear Relationship | 数值意义不明确是指在数学题目中，数值的描述不够清晰或具体，导致读者无法准确理解题意或进行正确的计算。这种错误通常发生在涉及比例、倍数或百分比的题目中，当数值的关系没有明确说明时，容易引起混淆。范例A. 鸡蛋有120个，鸭蛋比鸡蛋多(1/6)，请问两种蛋共有几个？B. 百家村要挖一条540米的水渠，第一天挖了20%，第二天挖了(1/8)，两天共挖了多少米？C. 商店去年的营业额是240万．相当今年的(4/5)，今年的营业额是多少？其中，A 描述数量关系时缺少'倍'字，B 缺少描述的主词而属于此类错误，正确的写法：多(1/6)倍、总长的(1/8)。而C，因为'相当今年的(4/5)' 就算不用'倍'字也知道它代表一个倍数，所以不算在此列。例如，全部的1/6, 总长的3/7, 某某的x/x, 题目单位不会有分数..等等，也不属于此错误。

(Unclear relationship refers to instances in mathematical problems where the description of the values is not sufficiently clear or specific, making it difficult for readers to accurately understand the problem or perform correct calculations. This type of error often occurs in problems involving ratios, multiples, or percentages, where the relationships between values are not clearly stated, leading to confusion.
Examples:
A. There are 120 chicken eggs, and the number of duck eggs is more than the chicken eggs by (1/6). How many eggs are there in total?
B. A village plans to dig a 540-meter canal. On the first day, they dug 20%, and on the second day, they dug (1/8). How many meters did they dig in total over the two days?
C. The store's revenue last year was 2.4 million, which is (4/5) of this year's revenue. What is this year's revenue in million units?
In example A, the description of the quantity relationship lacks the word "times" and thus falls into this error type of error; the correct phrasing should be "more by (1/6) times." In example B, the subject of the description is missing, making it another example of this error; the correct phrasing should be "1/8 of the total length." Example C, however, clearly states that "4/5" refers to a multiple of this year's revenue, so it does not fall under this error type. For instance, phrases like "1/6 of the total," "3/7 of the length," or "x/x of something," where units cannot be fractions, are also not considered this type of error.) | 数值意义不明确是指在数学题目中，数值的描述不够清晰或具体，导致读者无法准确理解题意或进行正确的计算。这种错误通常发生在涉及比例、倍数或百分比的题目中，当数值的关系没有明确说明时，容易引起混淆。具体情况包括：数值之间的关系不明确、缺少必要的数值说明、或数值的描述存在歧义。范例A. 鸡蛋有120个，鸭蛋比鸡蛋多(1/6)，请问两种蛋共有几个？B. 百家村要挖一条540米的水渠，第一天挖了20%，第二天挖了(1/8)，两天共挖了多少米？C. 商店去年的营业额是240万．相当今年的(4/5)，今年的营业额是多少万？D. 一桶啤酒倒出(2/3)，刚好倒出12千克．这桶啤酒原来重多少千克．E. 一列火车从上海开往天津，行了全程的(3/5)，距天津还有538千米．这列火车已经行了多少千米？其中，A 描述数量关系时缺少'倍'字，B 缺少描述的主词而属于此类错误，正确的写法：多(1/6)倍、总长的(1/8)。而C，因为'相当今年的(4/5)' 就算不用'倍'字也知道它代表一个倍数，所以不算在此列。例如，全部的1/6, 总长的3/7, 某某的x/x, 题目单位不会有分数..等等，也不属于此错误。D 属于此类，E 不属于此类

(Unclear relationship refers to situations in mathematical problems where the description of the values is not sufficiently clear or specific, leading to difficulties in accurately understanding the problem or performing correct calculations. This type of error often occurs in problems involving ratios, multiples, or percentages, where the relationships between values are not clearly stated, causing confusion. Specific issues include: unclear relationships between values, missing necessary numerical descriptions, or ambiguous descriptions of values.
Examples:
A. There are 120 chicken eggs, and the number of duck eggs is more by (1/6). How many eggs are there in total?
B. The village needs to dig a 540-meter canal. On the first day, they dug 20%, and on the second day, they dug (1/8). How many meters did they dig in total over the two days?
C. The store's revenue last year was 2.4 million, which is (4/5) of this year's revenue. What is this year's revenue in millions?
D. A barrel of beer is (2/3) empty, with 12 kilograms poured out. How much did the barrel originally weigh in kilograms?
E. A train travels from Shanghai to Tianjin, covering (3/5) of the total distance, with 538 kilometers remaining to Tianjin. How many kilometers has the train already traveled?
In example A, the description of the quantity relationship lacks the word "times," making it fall into this error type of error; the correct phrasing should be "more by (1/6) times." In example B, the subject of the description is missing, making it another example of this error; the correct phrasing should be "1/8 of the total length." Example C, however, clearly states that "4/5" refers to a multiple of this year's revenue, so it does not fall under this error type. Similarly, phrases like "1/6 of the total," "3/7 of the length," or "x/x of something," where units cannot be fractions, are also not considered this type of error. Examples D and E fall into this error type due to unclear numerical meanings.) |

Table 23: Example of prompt refinement for the calculation error type

| Error Type | Initial Definition and Example Generated by LLM | Definition Refined By LLM in the 2nd Round |
|---|---|---|
| Calculation Error | 不当数学符号是指在数学题目中使用了不正确或不明确的符号，导致题意模糊或误解。这种错误通常会使解题者无法正确理解题目要求，从而影响解题过程和结果。常见的情况包括缺少括号、使用错误的运算符号、或符号位置不当等。

范例A. 3-2/6的商，再加上5，结果=？ B. 1/2+1/3+1/4=？ C. 1/2+1/3+1/4=(( ())/(())) 其中，A 会有其他可能性(3-2/6的商+5) 属于此类错误，正确的写法：3-(2/6)+5。而B 为正常数学题目，不在此列。C 的括号使用为答案区，因此不在此列。

(Calculation error refer to the use of incorrect or unclear symbols in mathematical problems, leading to ambiguous or misunderstood problem statements. These errors can prevent the solver from correctly understanding the problem's requirements, thus affecting the problem-solving process and outcomes. Common issues include missing parentheses, incorrect use of operators, or improper placement of symbols. Examples:
A. The result of 3-2/6, plus 5, equals?
B. 1/2 + 1/3 + 1/4 = ?
C. 1/2 + 1/3 + 1/4 = ((())/(()))
In example A, there is ambiguity due to the placement of the symbols, potentially leading to multiple interpretations (3 - (2/6) + 5). The correct expression should be written as 3 - (2/6) + 5. Example B is a standard mathematical problem and does not fall into this error type. Example C uses parentheses to denote the answer area and therefore is not considered an error in symbol usage.) | 不当数学符号是指在数学题目中使用了不正确、不明确或不合适的符号，导致题意模糊或误解。这种错误通常会使解题者无法正确理解题目要求，从而影响解题过程和结果。常见的情况包括缺少括号、使用错误的运算符号、符号位置不当、符号不完整或符号格式错误等。

范例A. 3-2/6的商，再加上5，结果=？ B. 1/2+1/3+1/4=？ C. 1/2+1/3+1/4=(( ())/(())) D. (10/3)与(1/3)的和比(5/6)与(4/5)的和多多少？ E. 从(9/7)的倒数里减去(6/7)/(6/5)的商，差=？ 其中，A 会有其他可能性(3-2/6的商+5) 属于此类错误，正确的写法：3-(2/6)+5。而B 为正常数学题目，不在此列。C 的括号使用为答案区，因此不在此列。D 不属于此类，E 不属于此类。

(Calculation error refers to the use of incorrect, unclear, or inappropriate symbols in mathematical problems, resulting in ambiguous or misunderstood problem statements. These errors often prevent solvers from accurately understanding the problem's requirements, thereby affecting the problem-solving process and outcomes. Common issues include missing parentheses, incorrect use of operators, improper placement of symbols, incomplete symbols, or incorrect symbol formatting. Examples:
A. The result of 3-2/6, plus 5, equals?
B. 1/2 + 1/3 + 1/4 = ?
C. 1/2 + 1/3 + 1/4 = ((())/(()))
D. How much more is the sum of (10/3) and (1/3) compared to the sum of (5/6) and (4/5)?
E. The difference when subtracting the quotient of (6/7) divided by (6/5) from the reciprocal of (9/7) is?
In example A, there is ambiguity due to the placement of the symbols, potentially leading to multiple interpretations (3 - (2/6) + 5). The correct expression should be written as 3 - (2/6) + 5. Example B is a standard mathematical problem and does not fall into this error type. Example C uses parentheses to denote the answer area and therefore is not considered an error in symbol usage. Examples D and E are also not considered under this error type.) |

