# OpenReview forum: "Detecting Problematic Questions to Support Math Word Problem Design"
_ICLR.cc/2025/Conference — Submitted to ICLR 2025_

### Official Review · Reviewer_6cRD · 2024-11-04

**Soundness:** 2
**Presentation:** 2
**Contribution:** 1
**Rating:** 5
**Confidence:** 4

**Summary:**

The paper presents the MathError dataset, an extension of the Math23K dataset that includes error types for each question. The goal is to improve the clarity of questions within the dataset, and to achieve this, the authors define five key factors that form the basis for these error types. After annotating the questions based on these defined error types, the paper presents various prompting strategies that focus on self-optimization/reflection to evaluate how different LLMs perform in detecting these errors. The findings indicate that combining model-generated definitions of error types with human-provided examples can enhance the LLMs' performance in identifying the defined error types.

**Strengths:**

1. The research question studied in the paper is both interesting and important.
2. The experiments are thorough and cover various prompting scenarios and models.

**Weaknesses:**

1. The related work section is not thoroughly covered. Many studies have proposed various methods for self-reflection/evaluation, which were not mentioned in the paper.
2. Using only one annotator per batch of 5,815 makes the annotations prone to bias toward certain error types.
3. Although the paper addresses an interesting and important problem, I am not convinced that the proposed error types for improving clarity will achieve this effectively. Specifically, three out of the five items listed in Section 3 as factors for enhancing clarity seem to have the opposite effect:
- 3.1. Multiple Interpretations: When a question allows for different approaches to reach the final answer, it can actually make solving it easier for students. This flexibility increases their chances of finding a solution.
- 3.2. Informal Wording: Formal language does not always improve clarity. In the context of elementary school math questions, which the paper focuses on, using less formal language may help students understand the questions more easily.
- 3.3. Calculation Error: In the example provided on line 215, the phrasing “6000 divided by 59, and then subtract 35 from this result” is indeed difficult to interpret, as this form is uncommon. Additionally, it is unclear whether the question refers to regular division or floor division. By contrast, the original phrasing, "the difference between 6000/59 and 35," is far clearer and more familiar. Moreover, based on the paper, the goal of this factor is to address cases where "imprecise words are used to describe a mathematical expression." However, in this example, the mathematical operation is already there, so it is unclear what specific problem this factor intends to resolve.

  Given the aforementioned limitations in how clarity is evaluated, I believe the impact of this study on real-world applications may be minimal.

4. While the proposed method (PRO) is straightforward, the presentation in Section 5 is so awkward that it is impossible to understand the method without referring to the provided algorithm/figure.
5. Since the paper used the `GPT-4o` checkpoint, the results are likely not reproducible. This is because the `GPT-4o` checkpoint continues to receive updates and is not part of the isolated checkpoints designated for research purposes. Therefore, the results obtained may not be reliably reproduced.

**Questions:**

**Suggestion:**
1. Line 258: LLMs do not possess understanding or consciousness. Therefore, they do not have comprehension capability.

---

> ### Author Response · Authors · 2024-11-16
>
> - Response to W1:
>
> We acknowledge that there are many different approaches to self-reflection/evaluation in existing studies. In the current version, we have cited works most relevant to our research to ensure the focus remains on directly supporting our study. We may consider including additional discussions in the appendix of the final version. Thank you for your suggestion.
>
> - Response to W2:
>
> Thank you for your comment. While it is true that each batch of 5,815 samples was annotated by a single annotator, we designed an iterative process involving multiple rounds of feedback and discussion to ensure that annotators fully understood the task and maintained consistent labeling. This process included repeated annotation, review, and refinement, ultimately achieving substantial agreement.
>
>
> - Response to W3:
>
> Thank you for your feedback. However, allowing multiple interpretations or solutions in the context of math exam questions does not align with the primary goals of such assessments. Math questions are typically designed to have a single correct answer to ensure fairness and consistency in grading. If a question is ambiguous and allows for multiple interpretations, students may arrive at different answers due to varying understandings of the problem, which compromises the accuracy and reliability of the evaluation.
>
> Regarding the comment that "formal language does not always improve clarity," this does not apply to the context of formal assessments. Additionally, the "informal" we define does not refer to a more casual or easier-to-understand phrasing. Instead, it refers to errors arising from a lack of careful design in the problem's formulation, which can lead to confusion or misinterpretation. In formal tests, questions need to use precise and formal language to convey critical and important information. Using informal language could result in students misunderstanding the questions, which would undermine the fairness and validity of the assessment.
>
> For the Calculation Error issue, we understand that this may stem from challenges in effectively conveying the informality of the problem after translation into English. Specifically, as mentioned in Lines 127-129, Calculation Error refers to problems that use imprecise words to describe a mathematical expression, such as making it unclear whether to perform multiplication or division before addition or subtraction, which can lead to students calculating in the wrong order.
>
> As for the example, “6000 divided by 59, and then subtract 35 from this result,” this phrasing was intended to express the specific mathematical operation that the question should have conveyed. The original problem's description failed to provide this clarity. However, this is not meant to represent the final, formalized version of the question.
> We will address this more clearly in the final version. Thank you for your suggestion.
>
> - Response to W4:
>
> We understand your concerns about the potential real-world impact of this study. Sun et al. (2024) also explored similar issues. Building on their work, we conducted our study using a real-world dataset, further highlighting the importance of this problem.
> Besides, this work was inspired by a real-life experience where students encountered invalid math exam questions due to imprecise problem descriptions provided by teachers. These issues caused confusion for students and led to the questions being disqualified, with scores automatically compensated. This was a challenging situation for both students and teachers, highlighting the need for tools to improve question clarity.
>
> Sun, Y., Yin, Z., Guo, Q., Wu, J., Qiu, X., & Zhao, H. (2024, May). Benchmarking Hallucination in Large Language Models Based on Unanswerable Math Word Problem. In Proceedings of the 2024 Joint International Conference on Computational Linguistics, Language Resources and Evaluation (LREC-COLING 2024) (pp. 2178-2188).
>
>
> - W5: The presentation in Section 5 is so awkward.
>
> A5: Thank you for your feedback. We believe that the combination of the figure and text provides the most complete explanation of the proposed method. Regarding your comment on Section 5, if possible, we kindly ask for more specific suggestions for improvement, as the current feedback is somewhat discouraging without clear guidance.
>
> - W6:  Since the paper used the GPT-4o checkpoint, the results are likely not reproducible.
>
> A6: In Table 8, we also documented experiments using different sizes of Llama 3 models to explore the impact of model size on performance. We chose GPT-4o because it is one of the most powerful models currently available.
>
> - Q: Line 258: LLMs do not have comprehension capability.
>
> A: What we intended to convey is that the model processes task details based on patterns it has learned from data, rather than actual comprehension. We will rephrase this for clarity in the final version. Thank you

---

> ### Author Response · Authors · 2024-11-27
>
> Thank you again for your thoughtful comment. We would like to further elaborate on the data annotation process.
>
> The reason the remaining data was evenly divided among annotators (each annotating 5,815 samples) is due to the demanding nature of the annotation task. Given the challenge of comprehensively defining error types, it was crucial for us to work closely with annotators to ensure the initial definitions were appropriate and practical. This collaborative process required thorough discussions and refinement to align on the annotation criteria. And given the high precision required for this dataset, we had to carefully select highly responsible annotators with substantial knowledge and expertise.
> As a result, we could not delegate the task to a large pool of crowd workers, as it would have been difficult to ensure they fully understood and adhered to the specific requirements and nuances of the annotation task.
>
> However, if each data were annotated by multiple annotators, it would significantly increase their workload. This would not only place an immense burden on the annotators but also substantially raise the financial cost of the annotation process.
>
> Hence, this approach differs fundamentally from methods that rely on a large number of crowd workers. Instead, we worked with a small, trusted group of annotators. Our goal was to ensure the quality of the annotations while minimizing the burden on the annotators. To achieve this, we designed a process that prioritized both data quality and the well-being of the annotators.
>
> Finally, we validated the annotated data, which demonstrated a substantial level of consistency, achieving a Fleiss' kappa of 0.8103.

---

> ### Comment · Reviewer_6cRD · 2024-11-30
>
> I would like to thank the authors for their responses. However, my concerns remain unaddressed for the following reasons:
>
> > Response to W2
>
> Thank you for your response. However, the iterative process and discussion you described do not address the bias introduced by relying on a single annotator for labeling the data. Regarding the cost concerns you mentioned, while it is understandable to manage budgets, dataset creation is inherently a labor-intensive task. To ensure the reliability of the labeled dataset, it requires the involvement of at least a few annotators.
>
> > Response to W3
>
> With all due respect, I find the authors’ discussion on multiple interpretations unconvincing. Could you please provide references to support the claim that math questions are designed to have a single solution?
>
> Additionally, even if a question has multiple solutions, this does not necessarily mean it has different answers. A question can have multiple solutions while still leading to the same final answer. This approach offers more clarity for students, as it accommodates diverse ways of thinking. It allows more students to solve the question correctly using their unique perspectives. I do not understand why the authors have referred to this as "ambiguous."
>
> Regarding informality, could you please provide three distinct examples to clarify which factors this item is intended to improve?
>
> > W6: Since the paper used the GPT-4o checkpoint, the results are likely not reproducible.
>
> Thank you for the explanation. However, your response does not address my concern. My concern was not about model size comparison. What I highlighted is that each model typically has several isolated checkpoints, which stop receiving updates after a specific date. This date is usually reflected in the checkpoint’s name. The checkpoint you used, however, is not an isolated version and continues to receive updates, which makes it unsuitable for research purposes due to reproducibility.

---

> > ### Author Response · Authors · 2024-12-01
> >
> > 1. To ensure the reliability of the labeled dataset, it requires the involvement of at least a few annotators.
> >
> > A: We understand your concerns about potential biases from using a single annotator. To address this, we implemented measures such as rigorous training, iterative reviews, and consistency validation using a golden set. These efforts improved inter-annotator agreement from 0.6038 to 0.8103 (Fleiss' kappa), demonstrating strong alignment among annotators.
> > While we recognize that concerns may still remain, we have made every effort to balance practical constraints with rigorous standards. It is worth noting that similar approaches, involving a single annotator under specific circumstances, have been adopted in academic research. For example, as we explained in our response dated 28 Nov 2024, 11:32 (modified: 28 Nov 2024, 11:34)
> > https://openreview.net/forum?id=ma4SUzeCLR&noteId=UZvrETJFBu
> >
> >
> >
> >
> > 2. Questions about error types.
> >
> > A: We do not argue against the existence of multiple solution methods; in fact, encouraging diverse problem-solving approaches is beneficial for students. Our focus is on the issues of imprecise problem descriptions, one of which may lead students to interpret the question differently and arrive at varying answers—logically valid answers that may still diverge from the teacher’s intended question design. Such issues create unnecessary disputes over grading fairness.
> > As mentioned in our response on November 21, 2024, at 23:02, we emphasize the importance of having a single, definitive answer for assessment purposes, while still allowing flexibility in the methods used to arrive at that answer.
> > > In the Introduction (lines 31–35):
> > >
> > > The original price of an apple is 2 dollars. It has been discounted twice: the first discount is 10%, and the second discount is 5%. What is the current price of the apple?" It is unclear whether the second discount is to be applied to the original price or to the price after the first discount.
> > >
> > >In this case, the wording leads to two possible interpretations of the discount calculation: whether the second discount applies to the original price or to the price after the first discount. As a result, there could be at least two possible answers, which likely does not align with the teacher's intention. This question is primarily designed to test students' familiarity with multiplication operations, so having a single, definitive answer is crucial for quickly verifying whether students have mastered the skill, rather than allowing for open-ended responses.
> >
> > As presented in Table 2, some issues in problem descriptions, likely due to oversight by the question designers, result in missing information, such as the absence of "yuan" in "all books in the bookstore are priced at 0.95." While this type of issue may not lead to multiple answers, it reflects a lack of formality and precision. In formal exams, such omissions should be avoided, and our study seeks to address these shortcomings to improve the professionalism and clarity of question design.
> > In other words, we also aim to help teachers avoid careless mistakes in question formulation.
> >
> > I respectfully disagree with the notion that addressing the Calculation Error adds complexity. The original problem descriptions already suffer from unclear wording, which could confuse students. Our approach aims to resolve these uncertainties.
> >
> >
> >
> > 3. Not reproducible.
> >
> > A: We chose GPT-4o because it is one of the most powerful models currently available, which highlights the challenges inherent in this task. Even with such a capable LLM, we observed that the performance on certain aspects of the task remained suboptimal.
> > In Table 8, our experiments not only compared the performance of different model sizes but also tested additional settings, such as model-generated definitions versus human-written definitions. We observed that smaller parameter models performed poorly on this task, which led us to focus on larger models for more robust evaluation.
> >
> > We acknowledge the importance of reproducibility in research. To address this, we are willing to make the outputs of our methods publicly available, allowing others to compare their methods with ours. We believe this transparency will help the research community better assess our findings.
> > For now, the outputs were uploaded to an anonymous platform: https://anonymous.4open.science/r/PRO-54A0

---

> > > ### Comment · Reviewer_6cRD · 2024-12-03
> > >
> > > I would like to thank the authors for their responses. However, their replies mostly reiterate points that were already made. For example, regarding the ``GPT-4o`` checkpoint, the authors once again compared the model in terms of size and performance, which is entirely unrelated to the concern I raised. Therefore, I stand by my original score.

---

> > > > ### Author Response · Authors · 2024-12-03
> > > >
> > > > Thank you, Reviewer 6cRD, for your feedback. While we regret that our previous response did not address your concerns to your satisfaction, we would like to provide further clarification.
> > > >
> > > > 1. First, we would like to reiterate that there may have been some misunderstandings regarding the focus of our research. To address this, we supplemented our previous explanations with clarifications derived directly from the original arguments in our paper.
> > > >
> > > > 2. Regarding concerns about data annotation, we have made every effort to ensure its quality, following best practices and methodologies from related prior studies.
> > > >
> > > > 3. Lastly, in response to your concerns about the GPT-4o checkpoint, we want to emphasize that our discussion goes beyond merely comparing models of different sizes. We conducted experiments with varying settings, focusing on critical issues like error type definitions and whether examples should be generated by the model or authored by humans. Our findings indicate that smaller-scale models struggle to perform this task effectively, highlighting the inherent challenges of this problem. Consequently, we focused on larger models, such as GPT-3.5 and GPT-4o, for subsequent methodological development.
> > > >
> > > > Recent studies have also used GPT-3.5 and GPT-4 in experiments, and we provide below a list of papers from different teams exploring diverse topics. While their research areas differ from ours, these works demonstrate the value of conducting experiments with such models:
> > > >
> > > > * Jingwei Ni, Minjing Shi, Dominik Stammbach, Mrinmaya Sachan, Elliott Ash, and Markus Leippold. 2024. AFaCTA: Assisting the Annotation of Factual Claim Detection with Reliable LLM Annotators. In Proceedings of the 62nd Annual Meeting of the Association for Computational Linguistics (Volume 1: Long Papers), pages 1890–1912, Bangkok, Thailand. Association for Computational Linguistics.
> > > >
> > > > * Yuanyi Ren, Haoran Ye, Hanjun Fang, Xin Zhang, and Guojie Song. 2024. ValueBench: Towards Comprehensively Evaluating Value Orientations and Understanding of Large Language Models. In Proceedings of the 62nd Annual Meeting of the Association for Computational Linguistics (Volume 1: Long Papers), pages 2015–2040, Bangkok, Thailand. Association for Computational Linguistics.
> > > >
> > > > * Hiromasa Sakurai and Yusuke Miyao. 2024. Evaluating Intention Detection Capability of Large Language Models in Persuasive Dialogues. In Proceedings of the 62nd Annual Meeting of the Association for Computational Linguistics (Volume 1: Long Papers), pages 1635–1657, Bangkok, Thailand. Association for Computational Linguistics.
> > > >
> > > > To contribute to the research community, we have made our experimental results and model outputs publicly available, and we will release our code.

---

### Official Review · Reviewer_AN3L · 2024-11-04

**Soundness:** 3
**Presentation:** 3
**Contribution:** 2
**Rating:** 5
**Confidence:** 4

**Summary:**

The paper proposes a new, challenging, task for LLMs: identifying errors (and especially ambiguity) in math word problems which could be detrimental to learning for human students that try to solve them.
To facilitate this tasks, the authors annotate an existing dataset of MWPs in chinese language consisting of ca. 23k problems with five different error types.
The authors benchmarks various LLMs on the task of identifying such errors and propose a new method that uses iterative prompt optimization and can improve performance.

**Strengths:**

- The writing and presentation are clear
- The paper tackles and interesting, and challenging, new problem of identifying errors in generated math word problems

**Weaknesses:**

- Labeling 5,815 examples seems to be a very challenging and exhausting tasks for annotators, especially since none of the examples were also annotated by another annotator, as all of them received disjoint sets. This raises questions whether the annotators were still able to produce high-quality annotations and it would be important if the authors could clarify how this is ensured.
- The authors also mention that they are “unsure whether the five error types are sufficient” (L145) which might be due to them being established by manually inspecting only 200 examples. It would likely be helpful to either analyze more examples (as shown in L208 some errors only occur very infrequently) or to consult relevant literature, e.g. from the learning sciences.
- The self-optimization algorithm needs to be put better into perspective with related works.
- The experiments are a bit limited in scope, focussing only on proprietary LLMs (GPT-3.5, GPT-4o)

**Questions:**

- It would be good to back up statements like that ambiguity in questions needs to be reduced (which might be intuitive nevertheless) with literature from the learning sciences that supports them.
- L131: we focus extends -> our focus extends (?)
- Section 3 is titled: “From Math23k to MathError”. I would expect based on the title that it details how the authors construct MathError but the section only describes error types.
- The figure captions are rather short and could be improved, similarly Figure 2 does not have axis labels
- It would be helpful in Fig 1. a) to have a pointer where the start of the algorithm is
- The caption of Figure 2 suggests the y-axis shows the difference between refinement rounds but it actually shows the inverse (similarity)

---

> ### Author Response · Authors · 2024-11-16
>
> - Response to W1 and W2:
>
> As mentioned in Lines 140-204, we ensured the quality of annotations through an iterative process using a golden set of 100 samples with all error types. Annotators' performance was evaluated and refined over multiple rounds, ultimately reaching a Fleiss' kappa of 0.8103, indicating substantial agreement. Regarding the error types, annotators were encouraged to provide feedback and suggest refinements, making the definitions dynamically optimized during the process.
> During this annotation process, we aimed to ensure that all possible error types within this dataset were included. However, as noted in the limitations of our conclusion, this pilot study cannot define every possible error type that might exist. To address this, we propose the method described in Section 4 to construct the dataset.
>
> - Response to W3:
>
> We acknowledge that there are indeed many self-optimization algorithms. In the current version, we have cited the most relevant works. We will consider adding additional discussions in the appendix of the final version. Thank you for your comment.
>
> - Response to W4:
>
> Table 8 includes experiments that also test models of different sizes from Llama 3. We chose to use GPT-4o because it is one of the most powerful models currently available, and we wanted to showcase how a state-of-the-art model performs on this task.
>
>
> - Q1: Ambiguity reduction evidence needed.
>
> A1: Thank you for your comment. We agree with this point, and we have cited Sun et al. (2024). Besides, the motivation for this work stems from the author’s real-life experience. Specifically, we observed situations where students, during math exams, encountered issues caused by imprecise question descriptions written by teachers. This led to invalid questions and automatic score compensation. This personal experience underscores the importance of addressing ambiguity in math questions to ensure fairness and accuracy in assessments.
>
> Sun, Y., Yin, Z., Guo, Q., Wu, J., Qiu, X., & Zhao, H. (2024, May). Benchmarking Hallucination in Large Language Models Based on Unanswerable Math Word Problem. In Proceedings of the 2024 Joint International Conference on Computational Linguistics, Language Resources and Evaluation (LREC-COLING 2024) (pp. 2178-2188).
>
> - Q2: L131 suggestion
>
> A2: Thank you for pointing this out. We will correct it. We appreciate your feedback.
>
> - Q3: Section 3 is titled: “From Math23k to MathError”. I would expect based on the title that it details how the authors construct MathError but the section only describes error types.
>
> A3: Thank you for your comment. We structured the paper this way because we felt it was important to first explain the error types in Section 3 before moving on to Section 4, which details the dataset construction process. Section 4 specifically focuses on how MathError was constructed.
>
> - Q4: The figure captions are rather short and could be improved, similarly Figure 2 does not have axis labels
>
> A4: This is due to page limits. The detailed explanations are provided in the corresponding sections of the text. We can include additional details about Figure 2 in the final version.
>
> - Q5: It would be helpful in Fig 1. a) to have a pointer where the start of the algorithm is
>
> A5: We will add a pointer in the final version. Thank you.
>
> - Q6: The caption of Figure 2 suggests the y-axis shows the difference between refinement rounds but it actually shows the inverse (similarity)
>
> A6:  We used the term "difference" because the evaluation involves using the definitions from the previous round as a reference and the current round's definitions as candidates for comparison.

---

> > ### Comment · Reviewer_AN3L · 2024-11-27
> >
> > Dear Authors, thank you for your response and addressing most of my concerns, I have raised my score accordingly. However, my concern about annotator workload still remains, as is also point out by Reviewer 6cRD.

---

> ### Author Response · Authors · 2024-11-27
>
> Thank you for your feedback. However, we would still like to provide some additional clarification.
>
> "unsure whether the five error types are sufficient" means the inherent difficulty in comprehensively and perfectly defining all possible errors in mathematical problems. Our approach involves first sampling a subset of data for manual annotation to preliminarily identify a potential range of error types and define initial types. This process includes categorizing errors for each instance and determining whether certain defined types need to be merged. However, even with this effort, we cannot guarantee that all possible error types have been identified.
>
> To address this, we implemented an "Iterative Refinement Annotation" process, involving annotators directly in the annotation workflow. Through ongoing discussions with annotators, we ensured a shared understanding of the task while simultaneously refining the definitions of error types. As shown in our paper (Section 4), this iterative approach not only helped annotators better understand the task but also improved the quality of the annotated data.
>
> In Line 201, we also presented a second round of quality validation to ensure a high level of agreement among annotators regarding error types. The annotated data achieved substantial consistency with a Fleiss' kappa of 0.8103.
>
> Regarding why the remaining data was split among annotators (with each annotator responsible for 5,815 samples), it was due to the challenging nature of the annotation task. We needed to work very closely and carefully with the annotators to discuss and refine the definitions of error types, ensuring their completeness and accuracy. Additionally, we continuously monitored and supported the annotators to ensure the annotations were carried out correctly. Annotators were encouraged to reach out to us for discussions at any time, fostering an iterative and collaborative process. Given these requirements, it was difficult to delegate this annotation task to a larger pool of crowd workers, as it would have been difficult to ensure they fully understood and adhered to the specific needs of the project.
> If each data were annotated by multiple annotators, the workload would have been overly burdensome, and the costs significantly higher, especially given the precision required for this task. To mitigate this, we invited highly responsible annotators with substantial knowledge.
>
> We are aware of the concerns you mentioned regarding potential biases. Therefore, as explained in Lines 197–204, we detailed our quality evaluation process, including two rounds of validation, to demonstrate the high consistency of the annotated data. Under these circumstances, it was feasible to assign the remaining data to individual annotators, while maintaining close communication with them.
>
> This method differs fundamentally from the studies of inviting a large number of crowd workers. Instead, we worked with a small, trusted group of annotators who possess a solid foundation in mathematical knowledge. Additionally, we specifically sought annotators who were highly willing to collaborate closely with us, engaging in an iterative process of annotation and discussion to refine the dataset. To avoid overburdening them, we designed this process to ensure data quality while minimizing their workload.

---

### Official Review · Reviewer_ULVD · 2024-11-10

**Soundness:** 3
**Presentation:** 3
**Contribution:** 2
**Rating:** 6
**Confidence:** 4

**Summary:**

This paper introduces MathError, a dataset of annotated math word problems aimed at helping teachers identify and correct issues that could lead to ambiguous or unsolvable questions. By leveraging large language models (LLMs), the study demonstrates how these models can aid teachers in designing clear and consistent math problems. To do so, the authors propose a self-optimizing framework where the model iteratevily refines the demonstration.

**Strengths:**

- The paper is well written, has clear motivation, and has a well-explained methodology.

**Weaknesses:**

W1: How significant is the problem of erroneous questions? When humans communicate, language often contains implicit information and inherent ambiguity, yet people generally understand and perform well despite this. The main question is: how much do expression errors in math questions impact the performance of both humans and models? An analysis comparing the performance of large language models (LLMs) and humans on ambiguous math questions versus corrected versions would be valuable. Does performance improve when questions are clarified, or can models, like humans, effectively handle the natural ambiguity in language?

W2: The primary weakness of this paper’s proposed method is that the self-refinement technique performs poorly, with overall scores not exceeding 33% and some results falling below random chance. This may stem from ambiguities in defining error types or, as noted in W1, the model’s inability to detect "errors" that are subtle nuances of human language due to its inherent implicitness. How do humans perform on this task?

W3: Given that this work aims to address error detection, it is essential to include results on the model's overall error detection performance—specifically, how accurately it identifies the presence of any error. This should be supported with precision and recall scores to provide a clear measure of effectiveness.

**Questions:**

Q1: In the dataset construction phase, what is the background of the human annotators? Are they math experts? Please provide more details.

Q2: What is the distribution of the classes in the dataset? Are there descriptive statistics of the size of the dataset along with the number of samples per class?

Q3: Based on the results in Table 3, the authors state in Line 431 that model-generated definitions are better than humans. However, in Line 406, they state that based on Table 5, human-written definitions outperform the other prompting techniques. How are these results related? What is the setting presented in Table 3, then?

---

> ### Author Response · Authors · 2024-11-16
>
> - Response to W1:
>
> Thank you for your question, but it seems to diverge slightly from the core focus of our current research. Our primary goal is to detect errors in the descriptions of math problems to assist teachers in designing questions that accurately assess students' abilities. We focus on math problems with a single, definitive answer (aligned with the dataset’s original nature) rather than open-ended questions.
> In this context, the problem descriptions must be free from errors or imprecise expressions.
> Regarding your inquiry about analyzing LLMs solving problematic math questions, our experiment in Table 4 might provide some insights. When LLMs attempt to solve such imprecise problems, they may overlook potential issues in the problem descriptions, resulting in undetected errors, as discussed in Lines 393–394. This highlights that addressing such challenges remains a complex and open issue.
> Your suggestion about analyzing the impact of erroneous questions on the performance of humans and models is indeed an interesting direction. Thank you for the recommendation. We agree that such comparative analyses could be a valuable extension for future research.
>
> - Response to W2:
>
> The relatively low scores highlight the challenging nature of this issue, underscoring the value and necessity of conducting further research and exploration into this problem.
> Regarding W1’s point about the inherent implicitness of language, we emphasize again that this is irrelevant in this context, as math problems require precise descriptions.
> As for human performance on this task, Table 1 provides insights into this question. Initially, annotators might not have been fully familiar with the annotation tasks or the definitions of the error types. To address this, we developed a process where annotators gradually improved through multiple rounds of annotation and discussion (please refer to Lines 148–204). The results show that their performance in the final round was quite satisfactory.
>
> - Response to W3:
>
> Due to page limits, we did not include these results in this version. However, we fully agree on the importance of precision and recall metrics and will include them in the final version. Thank you for your suggestion.
>
> - Q1: In the dataset construction phase, what is the background of the human annotators? Are they math experts? Please provide more details.
>
> A1: The annotators were undergraduate students with academic backgrounds in fields related to mathematics or science. The math problems used in our study are at the elementary school level, which is within the scope of difficulty that these students can handle effectively.
> We briefly mentioned this in Lines 684-685 but provided limited information due to considerations for anonymity. In the final version, we will include more details on this aspect.
>
> - Q2: What is the distribution of the classes in the dataset?
>
> A2: Please refer to Lines 206-209 for this information.
>
>
> - Q3: How do the results in Table 3 and Table 5 relate?
>
> A3: In our experiments, we tested various methods and settings. Table 3 primarily presents the best settings under different methods. The results show that our proposed approach, using GPT-4o with human-written definitions and model-generated definitions, achieves promising performance.
> Table 5 focuses on evaluating several common prompting techniques (such as zero-shot, few-shot, and CoT promptings) for this task. Thus, we fixed one experimental variable by consistently using human-written definitions. The discussion in Table 5 is merely an observation of the experiment results and does not conflict with the conclusions drawn in Table 3.

---

> > ### Comment · Reviewer_ULVD · 2024-11-25
> >
> > Thank you for your response and explanations. Based on the new information, I have changed my score.

---

> > > ### Author Response · Authors · 2024-11-27
> > >
> > > Thank you for your positive feedback and acknowledgment of our work. We will carefully consider your suggestions and incorporate them into the final version of the paper.

---

### Author Response · Authors · 2024-11-21

We have noticed that there are some misunderstandings about flaws in math problem descriptions, such as "Multiple Interpretations" and "Informal Wording."
We would like to provide additional clarification here. For example, in the Introduction (lines 31–35):
> "The original price of an apple is 2 dollars. It has been discounted twice: the first discount is 10%, and the second discount is 5%. What is the current price of the apple?" It is unclear whether the second discount is to be applied to the original price or to the price after the first discount.

In this case, the wording leads to two possible interpretations of the discount calculation: whether the second discount applies to the original price or to the price after the first discount. As a result, there could be at least two possible answers, which likely does not align with the teacher's intention. This question is primarily designed to test students' familiarity with multiplication operations, so having a single, definitive answer is crucial for quickly verifying whether students have mastered the skill, rather than allowing for open-ended responses.

We consider this issue to be significant because the dataset we use consists of real exam questions. During the process of using this dataset, we identified numerous errors in problem descriptions, indicating a high likelihood of oversight during the problem design process.
Such oversights can lead to questions failing to effectively fulfill their intended educational or assessment purposes.

---

### Author Response · Authors · 2024-11-28
**Replying to Official Comment by Reviewer 6cRD and Reviewer AN3L**

We would like to thank the reviewers once again for their valuable comments, which have enabled us to add more details to help readers better understand our work.

We have already replied the concerns regarding the dataset construction in a separate response.
Below, we provide further clarification and additional details:

We chose to first establish a rigorous process to define error types and ensure that annotators thoroughly understood the task and reached a consensus. Following this, each annotator was assigned to label a specific portion of the data. This decision was guided by the nature of our research, which required annotators with a solid mathematical background and a willingness to engage closely with us to address any issues during the annotation process.

Given our limited budget, we adopted this annotation workflow to ensure the highest quality of labels while completing the dataset within the constraints of time and resources.

It is also worth noting that there are precedents in academic research where a single annotator has been employed for data labeling under specific circumstances. Therefore, our approach is not an uncommon practice in such cases:


* Gandhi, Nupoor, Anjalie Field, and Emma Strubell. "Annotating Mentions Alone Enables Efficient Domain Adaptation for Coreference Resolution." Proceedings of the 61st Annual Meeting of the Association for Computational Linguistics (Volume 1: Long Papers). 2023.

* Mehta, Maitrey, and Vivek Srikumar. "Verifying annotation agreement without multiple experts: A case study with Gujarati SNACS." Findings of the Association for Computational Linguistics: ACL 2023. 2023.

* Lin, Chin-Yi, et al. "Argument-Based Sentiment Analysis on Forward-Looking Statements." Findings of the Association for Computational Linguistics ACL 2024. 2024.

* Yen, An-Zi, Hen-Hsen Huang, and Hsin-Hsi Chen. "Personal knowledge base construction from text-based lifelogs." Proceedings of the 42nd International ACM SIGIR Conference on Research and Development in Information Retrieval. 2019.

---

### Meta-Review · Area_Chair_XhND · 2024-12-23

**Metareview:**

This paper introduces a dataset called MathError, which contains annotated math word problems, pointing out errors that could be an impediment for learning in human students.

Strengths:
The paper is well written, and well motivated.  The experimentation is solid.

Weaknesses:
Overall, I found that the reviewers are not extremely enthusiastic regarding the content of the paper--they point out reproducibility issues due to the utilization proprietary LLMs and issues with annotation methodology of utilizing one annotator for very large batches of data.  In addition, Reviewer 6cRD brings up a good point that they are not convinced that the proposed error types for improving clarity will achieve the overall goal effectively.

**Additional Comments On Reviewer Discussion:**

There is robust discussion between reviewers and authors that resulted in some score changes.

---

### Decision · Program_Chairs · 2025-01-22

Reject